# CLIBD: Bridging Vision and Genomics for Biodiversity Monitoring at Scale

**ZeMing Gong**[1]    **Austin T. Wang**[1]    **Xiaoliang Huo**[1]    **Joakim Bruslund Haurum**[2]
**Scott C. Lowe**[3]    **Graham W. Taylor**[3,4]    **Angel X. Chang**[1,5]
Simon Fraser University[1]    Aalborg University[2]    Vector Institute[3]
University of Guelph[4]    Alberta Machine Intelligence Institute (Amii)[5]
{zmgong, atw7, xiaoliang_huo, angelx}@sfu.ca, {joha}@create.aau.dk,
{scott.lowe}@vectorinstitute.ai, {gwtaylor}@uoguelph.ca
https://bioscan-ml.github.io/clibd/

## Abstract

Measuring biodiversity is crucial for understanding ecosystem health. While prior works have developed machine learning models for taxonomic classification of photographic images and DNA separately, in this work, we introduce a *multimodal* approach combining both, using CLIP-style contrastive learning to align images, barcode DNA, and text-based representations of taxonomic labels in a unified embedding space. This allows for accurate classification of both known and unknown insect species without task-specific fine-tuning, leveraging contrastive learning for the first time to fuse barcode DNA and image data. Our method surpasses previous single-modality approaches in accuracy by over 8% on zero-shot learning tasks, showcasing its effectiveness in biodiversity studies.

## 1 Introduction

As environmental change and habitat loss accelerate, monitoring biodiversity is crucial to understand and maintain the health of ecosystems. Taxonomic classification of organisms at scale is especially important for understanding regional biodiversity and studying species interactions.

To assist in this, researchers have used computer vision to identify organisms in images [22, 69, 72, 41] for a variety of applications such as ecological monitoring [13]. However, relying solely on images for identifying and classifying organisms fails to consider the rich evolutionary relationship between species and may miss fine-grained differences. To better capture these distinctions, researchers have used DNA sequences for genome understanding and taxonomic classification [31, 79, 45, 8, 2, 54]. In particular, *DNA barcodes* [26], short sections of DNA from specific genes such as the mitochondrial COI gene [40] in animals and ITS sequences in fungi, have been shown to be particularly useful for species identification [2, 54]. However, collecting DNA requires specialized equipment making it more expensive and less accessible than images. In this work, we investigate whether we can leverage recent advances in multi-modal representation learning [50, 32] to use information from DNA barcodes to guide the learning of image embeddings appropriate for taxonomic classification.

Recently, BioCLIP [61] used CLIP-style contrastive learning [50] to align images with common names and taxonomic descriptions to classify plants, animals, and fungi. While they showed that aligning image representations to text can help improve classification, taxonomic labels, which are not always available to the species level, are needed to obtain text descriptions. In this work, we study whether, by aligning to DNA barcodes (instead of text) during pretraining, we can learn improved representations of images for use in tasks relevant to biodiversity.

We propose CLIBD, which uses contrastive learning to map taxonomic labels, biological images and barcode DNA to the same embedding space. By leveraging DNA barcodes, we eliminate the reliance on manual taxonomic labels (as used for BioCLIP) while still incorporating rich taxonomic information into the representation. This is advantageous since DNA barcodes can be obtained at scale more readily than taxonomic labels, which require manual inspection from a human expert [23, 24, 60]. We also investigate leveraging partial taxonomic annotations, when available,

to build a trimodal latent space that aligns all three modalities for improved representations. We demonstrate the power of using DNA as a signal for aligning image embeddings by conducting experiments for fine-grained taxonomic classification down to the species level. Our experiments show our pretrained embeddings that align modalities can (1) improve on the representational power of image and DNA embeddings alone by obtaining higher taxonomic classification accuracy and (2) provide a bridge from image to DNA to enable image-to-DNA based retrieval.

## 2   RELATED WORK

We review work using images, DNA, and multi-modal models for fine-grained taxonomic classification of species and their application in biology. Prior work has primarily explored building unimodal models for either images or DNA, and largely relied on fine-tuning classifiers on a set of known species or higher-level taxa. This limits those approaches to a closed set of species, whereas we aim to also identify unseen species, for which we have no examples in the training set.

**Taxonomic classification of images in biology.** Many studies have explored image-based taxonomic classification of organisms [7, 69]. However, visual identification of species remains difficult due to the abundance of fine-grained classes and data imbalance among species. To improve fine-grained taxonomic classification, methods such as coarse and weak supervision [67, 53, 63] and contrastive learning [14, 74] have been developed. Despite these advances, image-based species classification is still limited, so we leverage DNA alongside images to enhance representation learning while maintaining the relative ease of acquiring visual data for new organisms.

**Representation learning for DNA.** Much work has focused on machine learning for DNA, such as for genome understanding [37, 34, 3, 35]. Recently, self-supervised learning has been used to develop foundation models on DNA, from masked-token prediction with transformers [31, 8, 65, 15, 79, 2], to contrastive learning [80] and next-character prediction with state-space models [47]. While much of this work focuses on human DNA, models have also been trained on large multi-species DNA datasets for taxonomic classification. BERTax [45] pretrained a BERT [18] model for hierarchical taxonomic classification but focused on coarser taxa like superkingdom, phylum, and genus, which are easier than fine-grained species classification. BarcodeBERT [2] showed that models pretrained on DNA barcodes rather than general DNA can be more effective for taxonomic classification. Though some of these works use contrastive learning, they do not align DNA with images. We extend these models by using cross-modal contrastive learning to align DNA and image embeddings, addressing the higher cost of obtaining DNA samples while improving image-based classification and enabling cross-modal queries.

**Multimodal models for biology.** While most work on taxonomic classification has been limited to single modalities, recent work started developing multimodal models for biological applications [29, 39, 77, 36]. Nguyen et al. [48] introduced Insect-1M, applying contrastive learning across text and image modalities. BioCLIP [61] pretrained multimodal contrastive models on images and text encodings of taxonomic labels in TreeOfLife-10M. However, these models focus only on *images and text*, limiting their use with new species where taxonomic labels are unavailable. They also miss leveraging the rich taxonomic knowledge from sources like the Barcode of Life Datasystem (BOLD), which at the time of writing has nearly 19 M validated DNA barcodes. Although many records include expert-assigned taxonomic labels, only 24% are labeled to the genus level and 9% to the species level in BOLD-derived datasets like BIOSCAN-1M and BIOSCAN-5M [23, 24]. By aligning images to DNA barcodes, we can use precise information in the DNA to align the image representations with the task of taxonomic classification, without requiring taxonomic labels.

One of the few works that uses both images and DNA is the Bayesian zero-shot learning (BZSL) approach by Badirli et al. [4]. This method models priors for image-based species classification by relating unseen species to nearby seen species in the DNA embedding space. Badirli et al. [5] similarly apply Bayesian techniques, with ridge regression to map image embeddings to the DNA space to predict genera for unseen species. However, this approach assumes prior knowledge of all genera and does not use taxonomic labels to learn its mapping, limiting its representational power. In this work, we show that aligning image and DNA modalities using end-to-end contrastive learning produces a more accurate model and useful representation space. By incorporating text during pretraining, we can leverage available taxonomic annotations without relying on their abundance.

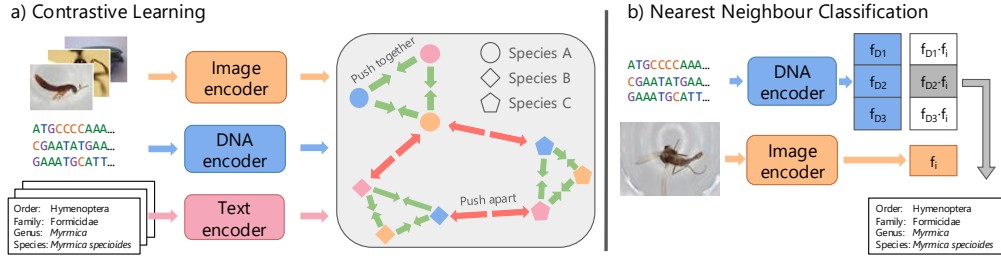

Figure 1: *Overview of CLIBD.* **(a)** Our model consists of three encoders for processing images, DNA barcodes, and text. During training, we use a contrastive loss to align the image, DNA, and text embeddings. **(b)** At inference, we embed a *query* image and match it to a database of existing image and DNA embeddings (*keys*). We use cosine similarity to find the closest key embedding and use its taxonomic label to classify the query.

Some prior work has investigated contrastive learning of image and DNA specifically for genetics and histology [64, 75, 44]. These works extend contrastive learning for aligning images and text—introduced by ConVIRT [78] for medical images and popularized by CLIP [50]—to images and human DNA. Here, we focus on multimodal contrastive learning with DNA barcodes and images for taxonomic classification. DNA barcodes (see Appendix D for background) are much shorter (400–800 base pairs) compared to sequences used in biomedical tasks (tens of thousands of base pairs).

## 3 METHOD

To align representations of images, DNA barcodes, and textual taxonomic labels, we start with a pretrained encoder for each modality and fine-tune them with a multimodal contrastive loss, illustrated in Figure 1. During inference, we use our fine-tuned encoders to extract features for a *query* image and match them against a database of image and DNA embeddings (*keys*) with known taxonomic labels. To classify a query image, we take the taxonomic labels associated with the most similar key. Whilst we can also query against the taxonomic text embeddings, this approach does not work well for taxa that were not seen during training. In contrast, our model can match queries against embeddings of labelled images and barcodes acquired after training. Thus, images and DNA barcodes comprise a more robust and comprehensive set of records against which to query.

### 3.1 TRAINING

**Contrastive learning.** We base our approach on a contrastive learning scheme similar to CLIP [50], which uses large-scale pretraining to learn joint embeddings of images and text. In contrastive learning, embeddings for paired specimens are pulled together while non-paired specimens are pushed apart, thus aligning the semantic spaces for cross-modal retrieval. Following prior work [55], we extend CLIP [50] to three modalities by considering each pair of modalities with the NT-Xent loss [58] between two modalities to align their representations. Let matrices $\mathbf{V}$, $\mathbf{D}$, and $\mathbf{T}$ represent the batch of $\ell_2$-normalized embeddings of the image, DNA, and text modalities. The $i$-th row of each matrix corresponds to the same physical specimen instance, thus rows $V_i$ and $D_i$ are image and DNA features from the same specimen, forming a positive pair. Features in different rows $V_i$ and $D_j$, $i \neq j$, come from different specimens and are negative pairs. The contrastive loss for pair $i$ is

$$L_i^{(V \to D)} = -\log \frac{\exp\left(V_i^T D_i / \tau\right)}{\sum_{j=1}^n \exp\left(V_i^T D_j / \tau\right)}, \qquad L_i^{(D \to V)} = -\log \frac{\exp\left(D_i^T V_i / \tau\right)}{\sum_{j=1}^n \exp\left(D_i^T V_j / \tau\right)},$$

where $\tau$ is a trainable temperature initialized to 0.07 following Radford et al. [50]. The total contrastive loss for a pair of modalities is the sum over the loss terms for each pairs of specimens,

$$L_{VD} = \sum_{i=1}^n \left( L_i^{(V \to D)} + L_i^{(D \to V)} \right),$$

wherein we apply the loss symmetrically to normalize over the possible paired embeddings for each modality [78, 55]. We repeat this for each pair of modalities and sum them to obtain the final loss, $L = L_{VD} + L_{DT} + L_{VT}$.

**Pretrained encoders.** For each modality we use a pretrained model to initialize our encoders. *Images:* ViT-B[1] pretrained on ImageNet-21k and fine-tuned on ImageNet-1k [21]. *DNA barcodes:*

---
[1]Loaded as `vit_base_patch16_224` in the timm library.

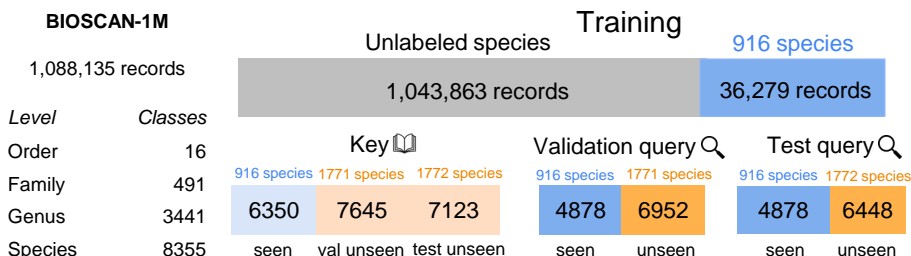

Figure 2: *Data partitioning.* We split the BIOSCAN-1M data into training, validation, and test partitions. The training set (used for contrastive learning) has records without any species labels as well as a set of *seen* species that are well-represented (at least 9 records per species). The validation and test sets include *seen* and *unseen* (not seen during training) species. These images are further split into subpartitions of *queries* (darker color) and *keys* (lighter color) for evaluation. We ensure that the validation and test sets have different *unseen* species. Since the *seen* species are common, we have a shared set of records (*Key*) that we use as *keys* for seen species and combine all key sets to form a reference database. We show the number of records in each box.

BarcodeBERT [2] with 5-mer tokenization, pretrained using masked language modelling on 893 k DNA barcodes [19] a dataset that is similar to but not overlapping with the DNA barcodes in the BIOSCAN-1M dataset, making it ideal for our study. *Text:* we use the pretrained BERT-Small [68] for taxonomic labels. Besides these pretrained encoders, we also conduct experiments using larger variants (see Appendix B.1.3).

## 3.2 INFERENCE

To use the model to predict taxonomic labels, we calculate the cosine similarity between the embedded input image (*query*) and reference image or DNA embeddings (*keys*) sampled from available species. We use the taxonomic label (order, family, genus, species) associated with the closest key as our prediction. This method allows us to evaluate the model in a zero-shot setting on species which were not seen by the model during training, provided we have appropriately labelled specimens to use as keys. The embedding space also provides the flexibility to be used for other downstream tasks, such as a supervised classifier or a Bayesian model [4, 5].

## 4 TASK AND DATA

To evaluate our method, we perform taxonomic classification using different combinations of input and reference modalities. The input may be a biological image or DNA sequence; this is matched against a reference set of labelled DNA barcodes, labelled biological images, or known taxonomic labels. We evaluate predictions at each taxonomic level by averaging accuracy over samples (micro) and taxon groups (macro). Unlike fine-tuning a species classification head, our approach can identify *unseen* species using labelled reference images or DNA, without needing to know all potential species at training time. We split the BIOSCAN-1M dataset so some species are "unseen" during training and report prediction accuracy for both seen and unseen species to study model generalization. This simulates the case where once a model is deployed, there are new specimens that are catalogued and labelled over time, which were not initially available for training the encoders.

**Dataset.** The BIOSCAN-1M dataset [23] is a curated collection of over one million insect data records sourced from a biodiversity monitoring workflow. Each record in the dataset includes a high-quality insect image, expert-annotated taxonomic label, and a DNA barcode. However, the dataset has incomplete taxonomic labels, with fewer than 10% of records labelled at the species level. This poses a challenge for conventional supervised methods, which require comprehensive species-level annotations, but our method is able to flexibly leverage partial or missing taxonomic information during contrastive learning. The dataset also possesses a long-tailed class imbalance, typical of real-world biological data. Given the vast biodiversity of insects—for which an estimated 80% is as-yet undescribed [62]—and the necessity to discern subtle visual differences, this dataset offers a significant challenge and opportunity for our model.

**Data partitioning.** We split BIOSCAN-1M into train/val/test sets to evaluate zero-shot classification and model generalization to unseen species. Records for well-represented species (at least 9 records) are partitioned at 80/20 ratio into *seen* and *unseen*, with seen records allocated to each of the splits and unseen records allocated to val and test. All records without species labels are used in

Table 1: Top-1 *macro*-accuracy (%) on BIOSCAN-1M *test* set for different combinations of modality alignment (image, DNA, text) during contrastive training. Results using DNA-to-DNA, image-to-image, and image-to-DNA query and key combinations. As a baseline, we also show results for unimodal pretrained models before cross-modal alignment and unimodal training using SimCLR (✓*). We report the accuracy for seen and unseen species, and their harmonic mean (H.M.) (**bold**: highest acc, *italic*: second highest acc.).

| Taxa | Aligned embeddings | | | DNA-to-DNA | | | Image-to-Image | | | Image-to-DNA | | |
|---|---|---|---|---|---|---|---|---|---|---|---|---|
| | Img | DNA | Txt | Seen | Unseen | H.M. | Seen | Unseen | H.M. | Seen | Unseen | H.M. |
| Order | ✗ | ✗ | ✗ | 78.8 | 91.8 | 84.8 | 54.9 | 48.0 | 51.2 | 7.7 | 9.6 | 8.5 |
| | ✓* | ✗ | ✗ | — | — | — | 69.5 | 59.8 | 64.3 | — | — | — |
| | ✓ | ✗ | ✓ | — | — | — | 99.6 | 97.4 | **98.5** | — | — | — |
| | ✓ | ✓ | ✗ | **100.0** | **100.0** | **100.0** | 89.5 | 97.6 | 93.4 | **99.7** | *71.8* | *83.5* |
| | ✓ | ✓ | ✓ | **100.0** | **100.0** | **100.0** | 99.7 | 94.4 | 97.0 | 99.4 | **88.5** | **93.6** |
| Family | ✗ | ✗ | ✗ | 86.2 | 82.1 | 84.1 | 28.1 | 21.7 | 24.5 | 0.5 | 0.8 | 0.6 |
| | ✓* | ✗ | ✗ | — | — | — | 43.6 | 30.9 | 36.2 | — | — | — |
| | ✓ | ✗ | ✓ | — | — | — | 90.7 | 76.7 | 83.1 | — | — | — |
| | ✓ | ✓ | ✗ | *99.1* | *97.6* | *98.3* | 89.1 | *81.1* | 84.9 | 90.2 | *44.6* | *59.7* |
| | ✓ | ✓ | ✓ | **100.0** | **98.3** | **99.1** | 90.9 | 81.8 | 86.1 | 90.8 | 50.1 | 64.6 |
| Genus | ✗ | ✗ | ✗ | 82.1 | 69.4 | 75.2 | 14.2 | 10.3 | 11.9 | 0.2 | 0.0 | 0.0 |
| | ✓* | ✗ | ✗ | — | — | — | 24.7 | 16.7 | 19.9 | — | — | — |
| | ✓ | ✗ | ✓ | — | — | — | 72.1 | 49.6 | 58.8 | — | — | — |
| | ✓ | ✓ | ✗ | *97.7* | *93.0* | *95.3* | 74.1 | 59.7 | 66.1 | **73.4** | *18.7* | *29.8* |
| | ✓ | ✓ | ✓ | **98.2** | **94.7** | **96.4** | 74.6 | 60.4 | 66.8 | *70.6* | 20.8 | 32.1 |
| Species | ✗ | ✗ | ✗ | 76.4 | 63.6 | 69.4 | 7.2 | 5.0 | 5.9 | 0.1 | 0.0 | 0.0 |
| | ✓* | ✗ | ✗ | — | — | — | 14.2 | 8.8 | 10.9 | — | — | — |
| | ✓ | ✗ | ✓ | — | — | — | 54.2 | 33.6 | 41.5 | — | — | — |
| | ✓ | ✓ | ✗ | *94.4* | *86.9* | *90.5* | 59.2 | **45.1** | 51.2 | 58.1 | *7.7* | *13.6* |
| | ✓ | ✓ | ✓ | **95.6** | **90.4** | **92.9** | 59.3 | 45.0 | 51.2 | *51.6* | 8.6 | 14.7 |

contrastive pretraining, and species with 2 to 8 records are divided between the *unseen* splits in the val and test sets. Importantly, we ensure that *unseen* species are mutually exclusive between the val and test sets and do not overlap with *seen* species for labelled records. Finally, among each of the seen and unseen sub-splits within the val and test sets, we allocate equal proportions of records as *queries*, to be used as inputs during evaluation, and *keys*, to be used as our reference database. Note that keys from seen and unseen species are combined to form the reference database. This procedure ensures we have a distribution of different cases in evaluation: both seen and unseen, and common and uncommon species. See Figure 2 for split statistics and Appendix A for details.

**Data preprocessing.** During inference, we resize images to 256×256 and apply a 224×224 center crop. For the DNA input, following Arias et al. [2], we set a maximum length of 660 for each sequence and tokenize the input into non-overlapping 5-mers. Similar to Stevens et al. [61], we concatenate the taxonomic levels of the insects together as text input. As we did not have the common names of each record, we used the Linnean order, family, genus, and species, up to the most specific available label per specimen. With this approach, we can still provide the model with knowledge of the higher-level taxonomy, even if some records do not have species-level annotations.

## 5 EXPERIMENTS

We evaluate the model's ability to retrieve correct taxonomic labels using images and DNA barcodes from the BIOSCAN-1M dataset [23]. This includes species that were either *seen* or *unseen* during contrastive learning. We also experiment on the INSECT dataset [4] for Bayesian zero-shot learning (BZSL) species-level image classification. We report the top-1 accuracy for the *seen* and *unseen* splits, as well as their harmonic mean (H.M.). We focus on evaluating the model using various combinations of modalities on the test set. Specifically, we assess the model's performance when using images and DNA barcodes as inputs, matched against their respective image and DNA reference sets, as well as the combination of image inputs matched to DNA references. In addition, we visualize the attention roll-out of the vision transformer we used as our image encoder to explore how the representation changes before and after contrastive learning and how aligning with different modalities affects the focus of the image encoder.

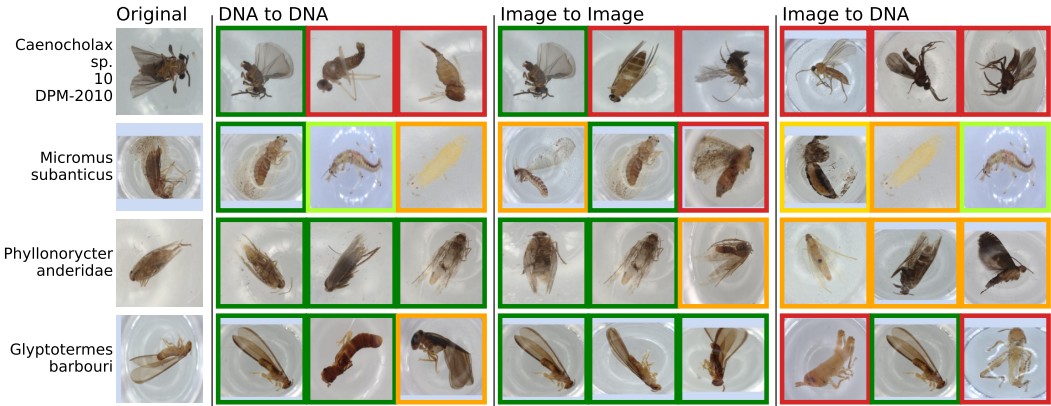

Figure 3: *Example query-key pairs.* Top-3 nearest specimens from the unseen validation-key dataset retrieved based on the cosine-similarity for DNA-to-DNA, image-to-image, and image-to-DNA retrieval. Box color indicates whether the retrieved samples had the same species (green), genus (light-green), family (yellow), or order (orange) as the query or, else not matched (red).

**Implementation details.** Models were trained on four 80GB A100 GPUs for 50 epochs with batch size 2000, using the Adam optimizer [33] and one-cycle learning rate schedule [57] with learning rate from $1e-6$ to $5e-5$. For efficient training, we use automatic mixed precision (AMP). We study the impact of AMP and batch size in Appendix C.

## 5.1 RETRIEVAL BY IMAGE AND DNA

We conducted experiments on BIOSCAN-1M [23] to study whether the accuracy of taxonomic classification improves with contrastive learning, particularly with the inclusion of DNA barcodes as an additional modality. We compare the accuracy of models trained to align different combinations of modalities: image (I), DNA (D), and text (T). We also assess performance by using different modalities as the query (input at inference time) and as the key (the embedding used for matching). As image is more readily available, we primarily focus on querying by image.

**Taxonomic classification**. In Table 1, we report the top-1 macro-accuracy on our BIOSCAN-1M test set for seen and unseen species (see Table 8 for the top-1 micro-accuracy). We report the performance of the different alignment models at different taxonomic levels (order, family, genus, species). As expected, the performance drops for more specific taxa (e.g., accuracy for order is much higher than for species), due to both the increased number of possible labels and the more fine-grained differences between them. When we consider unseen species, there is a drop in accuracy compared to seen species, suggesting there is a benefit to retraining encoders as more data is collected.

*Are multimodal aligned embeddings useful?* Our experiments show that by using contrastive learning to align images and DNA barcodes, we can 1) enable cross-modal querying and 2) improve the accuracy of our retrieval-based classifier. Unsurprisingly, we find that DNA-to-DNA retrieval is the most accurate, especially for species-level classification. Note that while direct DNA matching using methods like BLAST are quite effective, they tend to be slow and cannot be easily incorporated into neural networks for other downstream tasks [2]. By using contrastive learning to align different modalities, we enhance the image representation's ability to classify (image-to-image), especially at the genus and species level where the macro H.M. accuracy jumps from 12.5% to 69% (for genus) and 6.27% to 52% (for species) for our best model (I+D+T). Note that with alignment, the DNA-to-DNA retrieval performance also improves. To verify that the improvements in unimodal performance come from the flow of information between modalities, we also use unimodal contrastive loss following SimCLR [11] to fine-tune the image encoder on BIOSCAN-1M. Results (Table 1, second row) show that while the performance is better than the unaligned model, it is far from the gain we see in the I+D model (see Appendix B.1 for details).

*Do DNA barcodes provide a strong alignment target?* Table 1 also shows that on BIOSCAN-1M, using DNA provides a better alignment target than using taxonomic labels. Comparing the model that aligns image and text (I+T, row 3) vs. the one that aligns image and DNA (I+D, row 4), we see that the I+D model consistently gives higher accuracy than the I+T model. At the species level, the I+D can even outperform the I+D+T model. This is likely because the I+D+T model, despite using

Table 2: Species-level top-1 macro-accuracy (%) of BioCLIP and our CLIBD model on the test set, matching image embeddings (queries) against embeddings of different modalities for retrieval (image, DNA, and text keys). *Note:* the BioCLIP model [61] was trained on data that included BIOSCAN-1M but used different species splits, so it may have seen most of the unseen species during its training.

| | Aligned embeddings | | | Image-to-Image | | | Image-to-DNA | | | Image-to-Text | | |
|---|---|---|---|---|---|---|---|---|---|---|---|---|
| Model | Img | DNA | Txt | Seen | Unseen | H.M. | Seen | Unseen | H.M. | Seen | Unseen | H.M. |
| BioCLIP | ✓ | ✗ | ✓ | 20.4 | 14.8 | 17.1 | — | — | — | 4.2 | 3.1 | 3.6 |
| CLIBD | ✓ | ✗ | ✓ | 54.2 | 33.6 | 41.5 | — | — | — | *57.6* | *4.6* | *8.5* |
| CLIBD | ✓ | ✓ | ✓ | **59.3** | **45.0** | **51.2** | **51.6** | **8.6** | **14.7** | **56.0** | **4.8** | **8.9** |

all known labels for each sample, only has access to species labels for 3.36% of the pretraining data. While comprehensive species labels could yield a comparable model for taxonomic classification, the money and time cost for hiring expert scientific annotators makes it impractical to scale.

*Cross-modal retrieval.* Next we consider cross-modal retrieval performance from image to DNA. Without any alignment, image-to-DNA performance is effectively at chance accuracy, scoring extremely low for levels more fine-grained than order. By using contrastive learning to align image to DNA, we improve performance at all taxonomic ranks. While the cross-modal performance is still low compared to within-modal retrieval, we see that it is feasible to perform image-to-DNA retrieval, which unlocks the ability to classify taxa for which no images exist in reference databases.

**Retrieval examples.** Figure 3 shows examples of intra-modality image and DNA retrieval as well as image-to-DNA retrieval from our full model (aligning I+D+T), for which the retrieval is successful if the taxonomy of the retrieved key matches the image's. These examples show significant similarity between query and retrieved images across taxa, suggesting effective DNA and image embedding alignment despite differences in insect orientation and placement.

## 5.2 COMPARISON WITH BIOCLIP

Next we compare our aligned embedding space with that of BioCLIP [61] and investigate how well using taxonomic labels as keys would perform. We run experiments on BIOSCAN-1M by adapting the BioCLIP zero-shot learning demo script to perform species-level image classification. We use the BioCLIP pretrained model on the BIOSCAN-1M test set, with image query and either image or text embeddings as keys. For the text input for BioCLIP, we combined the four concatenated taxonomic levels with their provided `openai_templates` as text input, while for CLIBD, we used the concatenated labels only.

In Table 2, we report the species-level macro-accuracy. Results on other taxonomic levels (Table 13) follow a similar trend. We see CLIBD consistently outperforms BioCLIP, regardless of whether images or text is used as the key, and even for CLIBD trained only on images and text. Since BioCLIP was trained on a much broader dataset, including but not limited to BIOSCAN-1M, it may perform worse on insects as it was also trained on non-insect domains. CLIBD can also leverage DNA features during inference, while BioCLIP is limited to image and text modalities.

*Does matching images to taxonomic labels work better than matching to image or DNA embeddings?* The performance when using text as keys is much lower than using image or DNA keys. This shows that it is more useful to labeled samples with images (most preferred) or DNA. Nevertheless, if no such samples are available, it is possible to directly use text labels as matching keys.

## 5.3 ANALYSIS

**How does class size influence performance?** Since we use retrieval for taxonomic classification, it is expected that performance is linked to the number of records in the key set. Figure 4 confirms this. In general, accuracy is higher for seen species compared to unseen species in cross-modal retrieval, but this difference is less noticeable in within-modality retrieval. This suggests that contrastive training has better aligned the data it has been trained on, but it is less effective for unseen species. The DNA-DNA retrieval performance remains high, regardless of number of records in the key set.

**Attention visualization.** To investigate how the contrastive training changed the model, we visualize the attention roll-out of the vision transformer for the image encoder [1] in Figure 5. We reference the implementation method mentioned in Dosovitskiy et al. [21] by registering forward

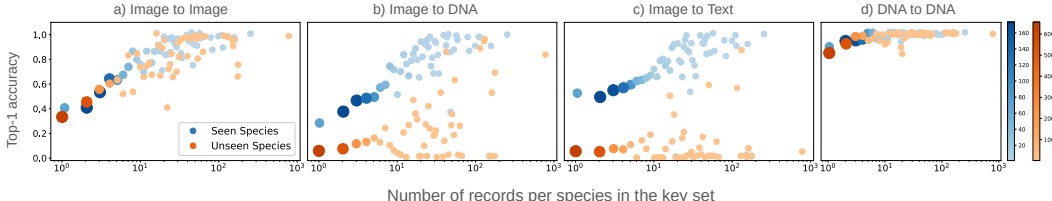

Figure 4: Average top-1 per-species accuracy, binned by count of species records in the key set, for different query and key combinations. We show *seen* in blue and *unseen* in orange, with size and color intensity indicating number of binned species. In all cases, the accuracy for seen species increases with the number of records. While the trend is similar for unseen species with intra-modal retrieval (*a*) and *d*)), cross-modal retrieval (*b*) and *c*)) achieve much lower performance, underscoring the challenge of cross-modal alignment.

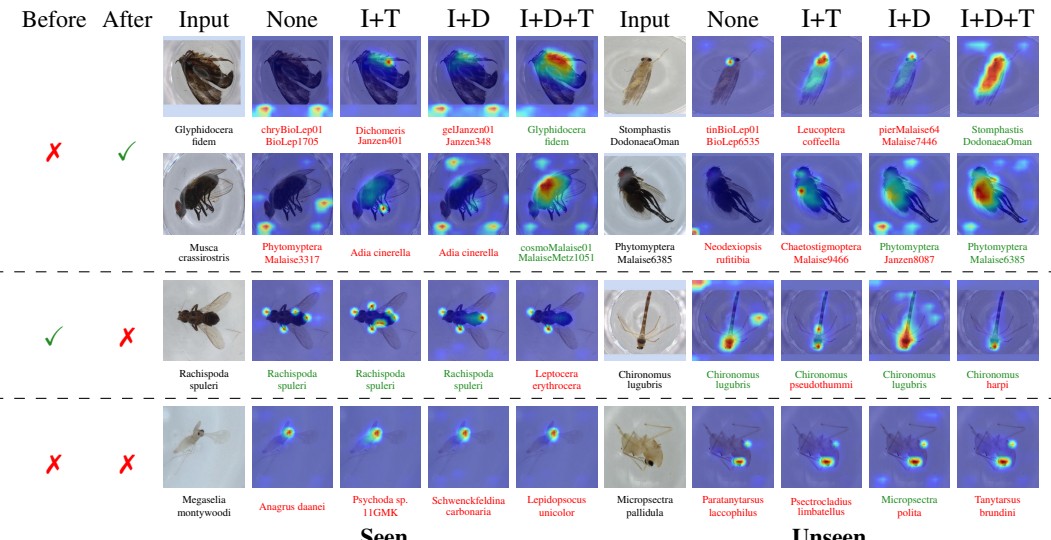

Figure 5: We visualize the attention for queries from *seen* and *unseen* species. The "Before" and "After" columns indicate if the prediction (at the species-level) was correct before (initial unaligned model) and after alignment (I+D+T model). Predicted genera + species are indicated in green for correct, red for incorrect. Only a few samples were predicted correctly before alignment and incorrect after (38 for seen and 69 for unseen).

hooks in the ViT's attention blocks to capture attention outputs and using the Rollout method to calculate attention accumulation. We then apply the processed mask to the original image to generate an attention map of the image area. Inspired by Darcet et al. [16], we inspect the mask of each attention block and remove attention maps containing artifacts. Ultimately, we select the forward outputs of the second to sixth attention blocks to generate the attention map.

We show examples for both seen and unseen species. For examples where the aligned models are able to predict correctly, we see that the attention is more clearly focused on the insect. Furthermore, we see that only the I+D+T model activates highly on the full insect, compared to I+T and I+D. Thus, while quantitatively the models may perform comparably, the I+D+T yields the most visually interpretable predictions. This is ideal for biodiversity, as interpretability is important for practical adoption. We also visualize the embedding space before and after alignment (see Appendix B.3) and show more examples of attention visualization (see Appendix B.4).

## 5.4 IMPROVING CROSS-MODAL CLASSIFICATION

We now more closely investigate how we can improve cross-modal ZS classification, where during inference time we have an image of an insect as a query, and we have a database of seen species (with images and DNA), and unseen species (with just DNA). Badirli et al. [4] proposed a hierarchical Bayesian model to classify images, using training images to learn the distribution priors and DNA embeddings to build surrogate priors for unseen classes. Here, we consider a similar zero-shot (ZS)

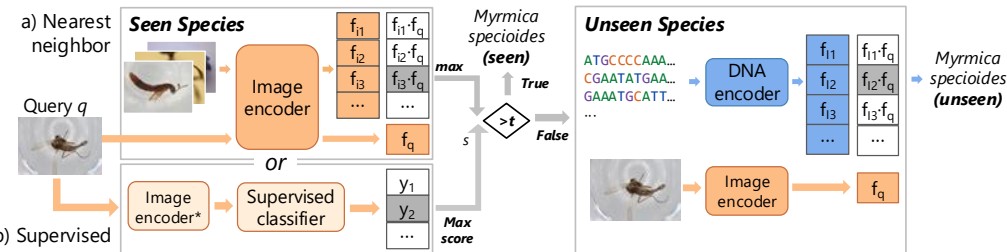

Figure 6: When we have DNA barcodes but not images of unseen species, we can use a combination of image and DNA as key sets. We adapt CLIBD for using images as keys for seen species and DNA for unseen species (i.e. the IS+DU strategy) to predict the species. We first classify the input image query $q$ against seen species, using either: (a) an 1-NN approach thresholding the cosine similarity score $s = \max_k f_{ik} \cdot f_q$; or (b) a supervised classifier predicting over all seen species and thresholding the maximum softmax probability $s = \max y_k$ by threshold $t$. If $s < t$, we subsequently query with the image feature $f_q$ using 1-NN with the DNA keys $f_{lk}$ of the unseen species and predict the unseen species of the closest DNA feature. *During supervised classifier training, the image encoder is finetuned only for use in the supervised pipeline.

Table 3: Top-1 accuracy (%) on our BIOSCAN-1M test set using the Image+DNA+Text model with image query. We compare nearest neighbour using only DNA keys (NN DNA), vs. our two strategies to use Image key for seen and DNA key for Unseen, either NN or a supervised linear classifier. We also compare against BZSL [4] with our embeddings.

| Taxa | Method | Strategy | Micro top-1 acc | | | Macro top-1 acc | | |
|------|--------|----------|------|--------|------|------|--------|------|
| | | | Seen | Unseen | H.M. | Seen | Unseen | H.M. |
| Genus | NN | DNA | **87.6** | 54.9 | **67.5** | 70.6 | 20.8 | **32.1** |
| | NN | IS+DU | 85.7 | 55.0 | 67.0 | 66.8 | 20.8 | 31.7 |
| | Linear | IS+DU | 83.6 | **55.6** | 66.8 | 61.4 | **21.1** | 31.5 |
| | BZSL | IS+DU | 86.8 | 46.5 | 60.6 | **75.7** | 14.4 | 24.2 |
| Species | NN | DNA | 74.2 | **27.8** | **40.4** | 51.6 | 8.6 | 14.7 |
| | NN | IS+DU | **76.1** | 26.2 | 39.0 | 54.8 | 8.5 | 14.8 |
| | Linear | IS+DU | 72.6 | 25.5 | 37.7 | 41.6 | **9.4** | **15.3** |
| | BZSL | IS+DU | **76.1** | 17.6 | 28.5 | **62.6** | 7.2 | 12.9 |

setting using our embeddings from our pretrained model for Bayesian zero-shot learning (BZSL), demonstrating its utility for unseen species classification.

We also consider a simpler strategy using the image embeddings for *seen* species, and DNA embeddings for *unseen* species. A two-stage approach first determines if a new image query represents a seen or unseen species (see Figure 6). For seen, image-to-image matching determines the species, while for unseen, image-to-DNA matching (assuming a reference set of labeled DNA samples for unseen species) determines the species. This is denoted by "IS-DU" (image seen - DNA unseen).

**Determining seen vs. unseen.** We frame the problem as an open-set recognition task [70] by using a classifier to determine whether an image query corresponds to a *seen* or an *unseen* species. This is useful for novel species detection as in practice we may not have a reference set of labeled samples or even set of species labels for unseen species. We compare using a 1-nearest neighbor (NN) classifier, and a linear supervised classifier with a fine-tuned image encoder (see Figure 6 left). See Appendix B.2 for details. We *show that without access to any unseen samples at inference time, our learned embeddings can be used to distinguish between seen and unseen* in Table 15. Using the image-to-image NN classifier, we obtain 83% accuracy on seen, 77% on unseen and a harmonic mean of 80%. For our linear classifier, we obtain lower accuracy on seen (73%), but higher on unseen (85%), with harmonic mean of 79%.

**Evaluation on BIOSCAN-1M.** Next we conduct experiments on BIOSCAN-1M to compare our IS+DU strategy vs. incorporating our learned embeddings in BZSL. We also compare against querying the seen and unseen DNA keys using 1-NN directly. Table 3 reports the top-1 micro and macro accuracy of at the genus and species level (see Table 16 for order and family level classification). We find that at the genus and species level, BZSL obtains good performance for seen species but that our simple NN-based approach actually outperforms BZSL on unseen species. Using our IS-DU strategy with the supervised linear classifier, we obtain the best macro top-1 accuracy on unseen species, demonstrating that *the complexity of BZSL may not be necessary*.

Table 4: Macro-accuracy (%) for species classification in a Bayesian zero-shot learning task on the INSECT dataset. We compare our CLIBD-**D** with several DNA encoders: CNN encoder [4], DNABERT-2 [79], BarcodeBERT [2]. The baseline image encoder ResNet-101 used in Badirli et al. [4] is compared against our image encoder before (ViT-B) and after (CLIBD-**I**) pretraining on BIOSCAN-1M (BS-1M). We indicate the pretraining set for DNA (Pre-DNA) as the multi-species (M.S.) set from Zhou et al. [79], anthropods from Arias et al. [2], or BS-1M. We compare models both with and without supervised fine-tuning (FT) for each encoder, except for CLIBD-**D**, where the comparison is with or without contrastive learning for fine-tuning on the INSECT dataset. We highlight the baseline from Badirli et al. [4] and the variant with our fine-tuned encoders in gray.

| DNA enc. | Image enc. | Data sources | | | Species-level acc (%) | | |
|---|---|---|---|---|---|---|---|
| | | Pre-DNA | FT-DNA | FT-Img | Seen | Unseen | H.M. |
| CNN encoder | RN-101 | – | INSECT | – | 38.3 | 20.8 | 27.0 |
| DNABERT-2 | RN-101 | M.S. | – | – | 36.2 | 10.4 | 16.2 |
| DNABERT-2 | RN-101 | M.S. | INSECT | – | 30.8 | 8.6 | 13.4 |
| BarcodeBERT | RN-101 | Arthro | – | – | 38.4 | 16.5 | 23.1 |
| BarcodeBERT | RN-101 | Arthro | INSECT | – | 37.3 | 20.8 | 26.7 |
| BarcodeBERT | ViT-B | Arthro | INSECT | – | 42.4 | 23.5 | 30.2 |
| BarcodeBERT | ViT-B | Arthro | INSECT | INSECT | 54.1 | 20.1 | 29.3 |
| CNN encoder | CLIBD-**I** | – | INSECT | – | 37.7 | 16.0 | 22.5 |
| BarcodeBERT | CLIBD-**I** | Arthro | INSECT | – | 52.0 | 21.6 | 30.6 |
| BarcodeBERT | CLIBD-**I** | Arthro | INSECT | INSECT | 48.5 | 23.0 | 31.2 |
| CLIBD-**D** | RN-101 | BS-1M | – | – | 34.7 | 21.3 | 26.4 |
| CLIBD-**D** | RN-101 | BS-1M | INSECT | – | 32.8 | 25.0 | 28.4 |
| CLIBD-**D** | CLIBD-**I** | BS-1M | – | – | 34.2 | 22.1 | 26.9 |
| CLIBD-**D** | CLIBD-**I** | BS-1M | INSECT | INSECT | **57.9** | **25.1** | **35.0** |

**Evaluation on INSECT dataset with BZSL.** In this experiment, we study whether our image and DNA embeddings can improve performance when incorporated into BZSL. We evaluate on the INSECT dataset [4], which contains 21,212 pairs of insect images and DNA barcodes from 1,213 species. We record the performance of different combinations of encoders. We start with the original CNN-based DNA encoder and ResNet-101 image encoder pretrained on ImageNet-1K from Badirli et al. [4]. We also consider ViT-B [21], pretrained on ImageNet-21k and fine-tuned on ImageNet-1k. For DNA encoders, we compare against BarcodeBERT [2] (which was used to initialize our CLIBD model), and DNABERT-2 [79], a BERT-based model trained on multi-species DNA.

Table 4 shows that *incorporating CLIBD improves BZSL accuracy*. With the baseline image encoder (RN-101), CLIBD-**D** performs the best of the DNA encoders (both with and without DNA fine-tuning). With BarcodeBERT, our CLIBD-**I** outperforms both RN-101 and ViT-B. CLIBD-**I** together with CLIBD-**D** gives the best performance with accuracy after fine-tuning of 57.9% for seen and 25.1% for unseen. This shows the benefits of using CLIBD to learn a shared embedding space relating image and DNA data, both in performance and flexibility of applying to downstream tasks.

# 6   CONCLUSION

We introduced CLIBD, an approach for integrating biological images with DNA barcodes and taxonomic labels to improve taxonomic classification. It uses contrastive learning to align embeddings in a shared latent space. Our experiments show that, using barcode DNA as an alignment target for image representations, CLIBD outperforms models that only align images and text. Pretraining with text-encoded taxonomic labels, even when scarce at fine-grained levels, further improves performance. We further demonstrate the effectiveness of our aligned embeddings in zero-shot image-to-DNA retrieval. Our BIOSCAN-1M and INSECT dataset experiments showcase the value of using barcode DNA as an alignment target. However, the availability of barcode DNA is limited (see Appendix D for discussion) compared to human-curated image datasets with taxonomic labels such as TreeOfLife-10M [61] and BioTrove [76]. It would be interesting to couple the rich information found in DNA with these datasets that include natural images, and go beyond insects. It is also possible to investigate other multi-modal learning schemes beyond CLIP. Overall, we believe that using DNA barcodes for improved representation of organism images is a promising avenue for future research. We hope that our work can stimulate the application of multimodal machine learning with barcode DNA to help build improved automated tools for biodiversity monitoring.

## ACKNOWLEDGEMENT

We acknowledge the support of the Government of Canada's New Frontiers in Research Fund (NFRF) [NFRFT-2020-00073]. AXC and GWT are also supported by Canada CIFAR AI Chair grants. JBH is supported by the Pioneer Centre for AI (DNRF grant number P1). This research was enabled in part by support provided by the Digital Research Alliance of Canada (alliancecan.ca). We also thank Mrinal Goshalia and Kian Hosseinkhani for testing the code, Han-Hung Lee and Yiming Zhang for helpful discussions, and Manolis Savva for proofreading and feedback.

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

## APPENDICES

We provide additional details on how we obtain our data split (Appendix A), additional results (Appendix B) on the validation set, using different image and text encoders, additional details about the cross-modal experiments, and visualize the embedding space. We also include experiments with hyperparameter settings such as the use of automatic mixed precision and batch size (Appendix C). In Appendix D, we provide additional information on the utility and cost of DNA barcodes.

## A  ADDITIONAL DATA DETAILS

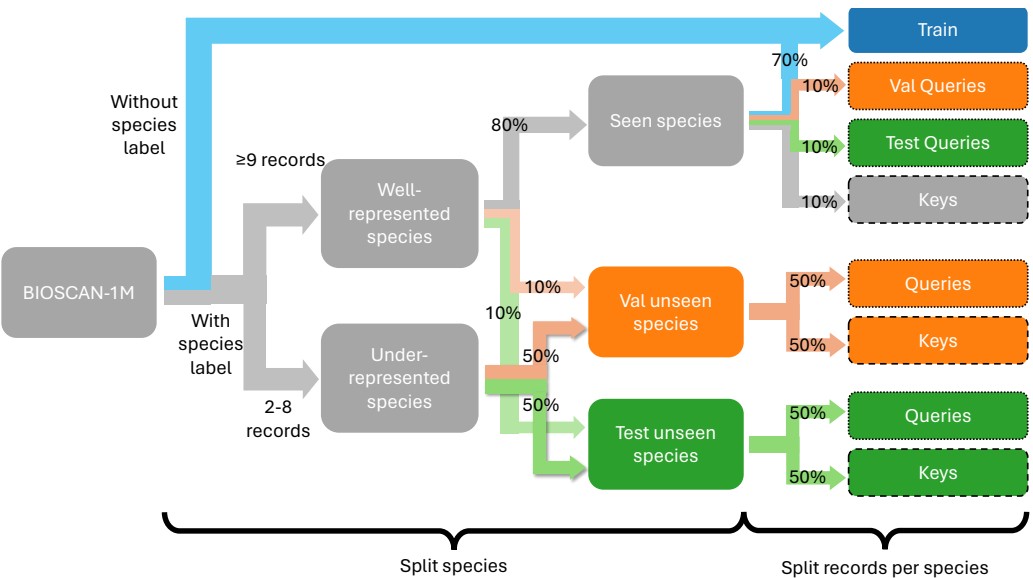

Figure 7: *Data partitioning strategy*. We first partition species among the splits based on the presence of a species label and the number of records per species, and then each species is designated as seen or unseen. Records from each species are then partitioned among train (blue), validation (orange), and test (green). For the validation and test sets, some records are used as *queries*, and the rest are used as *keys* for the reference database for retrieval.

**Data partitioning process.** We use a multi-stage process to establish our split of BIOSCAN-1M [23] for our experiments (see Figure 7). Firstly, we separate records with and without species labels. Any record without a species label is allocated for pretraining, as we cannot easily use them during evaluation. Of the remaining records with labelled species, we partition species based on their number of samples. Species with at least 9 records are allocated 80/20 to *seen* and *unseen*, with unseen records split evenly between validation and test. Species with 2 to 8 records are used only as unseen species, with a partition of 50/50 between validation and test. This allows us to simulate real-world scenarios, in which most of our unseen species are represented only by a few records, ensuring a realistic distribution of species sets. Species with only one record are excluded, as we need at least one record each to act the query and the key, respectively.

Finally, we allocate the records within each species into designated partitions. For the seen species, we subdivide the records at a 70/10/10/10 ratio into train/val/test/key, where the keys for the seen species are shared across all splits. The unseen species for each of validation and test are split evenly between queries and keys. The allocation of queries and keys ensures that we have clearly designated samples as inputs and target references for inference. We note that some samples in our data may have the exact same barcode even though the image may differ.

**Data statistics.** As a result of this process, we obtain a seen split with 916 species, and unseen val and test splits with 1771 and and 1772 species respectively (see Figure 2 in the main paper). We present the number of species and the distribution of records across species (Figure 8) and the average number of records per taxa level (Table 5). In Figure 8, we show the number of records for

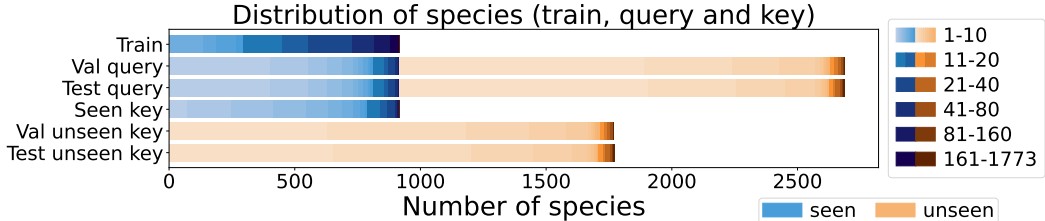

Figure 8: *Distribution of number of records per species* for our BIOSCAN-1M train, query and key splits. We show the seen and unseen species in blue and orange, with the intensity of the color indicating the number of records for that species. Note that the seen species are shared across train, val, test while the val and test unseen species are distinct from each other. From the distribution, we see that most of the species in the unseen splits have fewer than 10 records (light orange).

Table 5: *Data statistics.* Average number of records per taxa level. Note that the seen and unseen species are distinct, but there is overlap at other levels of the taxa (order, family, genus) between seen and unseen species.

|  | | Species seen | | | Species unseen | | | |
| --- | --- | --- | --- | --- | --- | --- | --- | --- |
|  | Train | Key | Val Query | Test Query | Val Key | Val Query | Test Key | Test Query |
| order | 60007.72 | 634.80 | 487.60 | 487.60 | 545.71 | 496.29 | 593.58 | 537.33 |
| family | 2305.88 | 41.75 | 33.74 | 33.74 | 38.99 | 36.34 | 34.89 | 31.25 |
| genus | 85.63 | 6.10 | 5.17 | 5.17 | 4.98 | 4.73 | 4.56 | 4.36 |
| species | 21.79 | 2.06 | 1.99 | 1.99 | 1.92 | 1.92 | 1.70 | 1.69 |

each species by color intensity, with lighter hues for species with fewer records and darker hues for species with more records. Due to the long-tailed distribution of species, there are just a few species with a large number of records, and most species with just a few records. This is especially true for the unseen split, where 90% of species have 4 or less records. Similarly, we see from Table 5 that each species has an average of only 1 to 2 samples in the query set or the key set.

In Table 6, we present the number of classes at different taxonomy levels. Specifically, we show the overlap of classes between the seen and unseen sets at various levels. We separate the statistics for records that do not have any species labels (which go into the pretrain split), and the those that are used to establish the train/val/test splits. Note that in the pretrain split, the statistics are lower bounds at the genus and species level as records are missing the species label and potential genus label as well. Due to our data division method, there is no overlap at the species level. However, for higher taxonomic levels there are overlaps. As the taxonomy level increases from genus to order, the proportion of overlapping classes gradually increases, representing a larger percentage of the total number of classes at each level.

**Task difficulty.** To understand the difficulty of classification at different levels of the taxonomy, we consider the performance of a simple majority baseline classifier. In Table 7, we present the micro

Table 6: *Taxa overlap between seen and unseen species.* Overlap at different levels of the taxa (order, family, genus) between seen and unseen species. We separately report the statistics for records without species labels (which form the pretrain set), and those that we used to build our train/val/test set. Note that for the pretrain split, the number of classes at the genus and species level are lower bounds, as some labels are missing. Gray highlight indicate number of classes at each taxonomy level for either seen or unseen. Cyan highlight indicate the number of classes that overlap at that level across seen / unseen.

|  | | Pretrain (no species) | | | Train/val/test | | |
| --- | --- | --- | --- | --- | --- | --- | --- |
|  | total | seen | unseen | overlap | seen | unseen | overlap |
| order | 19 | 10 | 13 | 10 | 10 | 14 | 10 |
| family | 479 | 129 | 219 | 112 | 132 | 239 | 113 |
| genus | 2731 | 351 | 686 | 234 | 521 | 1400 | 279 |
| species | 4459 | 0 | 0 | 0 | 916 | 3543 | 0 |

Table 7: *Chance performance.* Performance for most frequent class baseline for micro-accuracy and chance performance (random class) for macro-accuracy (%).

| | Micro | | | | | | Macro | | | | |
|--|--|--|--|--|--|--|--|--|--|--|--|
| | Seen | Unseen | | H.M. | | Seen | Unseen | | H.M. | |
| | val/test | val | test | val | test | val/test | val | test | val | test |
| Order | 76.32 | 58.46 | 57.27 | 66.21 | 65.44 | 10.00 | 7.14 | 8.33 | 9.09 | 7.69 |
| Family | 15.33 | 24.64 | 19.12 | 18.90 | 17.02 | 0.76 | 0.58 | 0.55 | 0.63 | 0.56 |
| Genus | 7.46 | 24.17 | 18.55 | 11.40 | 10.64 | 0.19 | 0.12 | 0.12 | 0.14 | 0.12 |
| Species | 5.21 | 19.52 | 12.14 | 8.22 | 7.29 | 0.11 | 0.06 | 0.06 | 0.07 | 0.06 |

and macro accuracy when the most frequent class at each taxonomy level is chosen for prediction. The macro accuracy is the same as chance performance of selecting a class at random, while the micro accuracy depends on the fraction of records with the most frequent class. The significant difference between micro and macro accuracy indicates the large class imbalance in our dataset. This imbalance reflects the phenomenon that some categories are more common than others in nature as well as the sampling bias present in data collection efforts.

# B  ADDITIONAL EXPERIMENTS

In this section, we include additional experimental results and visualizations. We provide additional results on BIOSCAN-1M (Appendix B.1) and image to DNA retrieval results (Appendix B.2). We also visualize the aligned embedding space (Appendix B.3) to show the model's capability in integrating and representing diverse biological data, and more attention visualizations (Appendix B.4).

## B.1  ADDITIONAL CLASSIFICATION RESULTS ON BIOSCAN-1M

Table 8: Top-1 *micro* accuracy (%) on our *test* set for BIOSCAN-1M and different combinations of aligned embeddings (image, DNA, text) during contrastive training. We show results for using image-to-image, DNA-to-DNA, and image-to-DNA query and key combinations. As a baseline, we show the results prior to contrastive learning (uni-modal pretrained models without cross-modal alignment). We report the accuracy for seen and unseen species, and the harmonic mean (H.M.) between these (**bold**: highest acc, *italic*: second highest acc.). We also compare against unimodal training using SimCLR (2nd row in each taxa, indicated with ✓*).

| Taxa | Aligned embeddings | | | DNA-to-DNA | | | Image-to-Image | | | Image-to-DNA | | |
|--|--|--|--|--|--|--|--|--|--|--|--|--|
| | Img | DNA | Txt | Seen | Unseen | H.M. | Seen | Unseen | H.M. | Seen | Unseen | H.M. |
| Order | ✗ | ✗ | ✗ | 99.1 | 98.5 | 98.8 | 88.8 | 90.8 | 89.8 | 10.5 | 11.0 | 10.7 |
| | ✓* | ✗ | ✗ | — | — | — | 94.4 | 94.1 | 94.3 | — | — | — |
| | ✓ | ✗ | ✓ | — | — | — | **99.7** | 99.6 | **99.6** | — | — | — |
| | ✓ | ✓ | ✗ | **100.0** | **100.0** | **100.0** | 99.6 | **99.7** | **99.6** | 99.7 | *98.9* | *99.3* |
| | ✓ | ✓ | ✓ | **100.0** | **100.0** | **100.0** | **99.7** | 99.6 | **99.6** | 99.7 | **99.3** | **99.5** |
| Family | ✗ | ✗ | ✗ | 96.2 | 93.8 | 95.0 | 52.9 | 60.0 | 56.2 | 1.0 | 1.1 | 1.0 |
| | ✓* | ✗ | ✗ | — | — | — | 67.5 | 69.7 | 68.6 | — | — | — |
| | ✓ | ✗ | ✓ | — | — | — | 95.7 | 92.2 | 93.9 | — | — | — |
| | ✓ | ✓ | ✗ | 99.8 | 99.2 | 99.5 | 95.9 | *93.1* | 94.5 | 95.8 | 84.6 | 89.9 |
| | ✓ | ✓ | ✓ | **100.0** | **99.5** | **99.7** | **96.2** | **93.7** | **94.9** | 96.5 | 87.1 | 91.6 |
| Genus | ✗ | ✗ | ✗ | 93.4 | 89.0 | 91.1 | 30.1 | 38.7 | 33.9 | 0.2 | 0.1 | 0.1 |
| | ✓* | ✗ | ✗ | — | — | — | 43.2 | 48.9 | 45.9 | — | — | — |
| | ✓ | ✗ | ✓ | — | — | — | 87.2 | 77.1 | 81.8 | — | — | — |
| | ✓ | ✓ | ✗ | 99.2 | 96.9 | 98.0 | 88.6 | *82.1* | 85.2 | **87.8** | *51.3* | 64.8 |
| | ✓ | ✓ | ✓ | **99.5** | **97.9** | **98.7** | **89.3** | **82.3** | **85.7** | 87.6 | **54.9** | **67.5** |
| Species | ✗ | ✗ | ✗ | 90.4 | 84.6 | 87.4 | 18.1 | 26.8 | 21.6 | 0.1 | 0.1 | 0.1 |
| | ✓* | ✗ | ✗ | — | — | — | 28.5 | 36.2 | 31.9 | — | — | — |
| | ✓ | ✗ | ✓ | — | — | — | 76.2 | 61.9 | 68.3 | — | — | — |
| | ✓ | ✓ | ✗ | 97.9 | *94.8* | 96.3 | 79.2 | **70.0** | **74.3** | **75.1** | 25.2 | *37.7* |
| | ✓ | ✓ | ✓ | **98.4** | **96.3** | **97.3** | **79.6** | *69.7* | **74.3** | 74.2 | **27.8** | **40.4** |

Table 9: Top-1 *macro* accuracy (%) on our *val* set for BIOSCAN-1M and different combinations of aligned embeddings (image, DNA, text) during contrastive training. We show results for using image-to-image, DNA-to-DNA, and image-to-DNA query and key combinations. As a baseline, we show the results prior to contrastive learning (uni-modal pretrained models without cross-modal alignment), and unimodal training using SimCLR (✓*). We report the accuracy for seen and unseen species, and the harmonic mean (H.M.) between these (**bold**: highest acc, *italic*: second highest acc.).

| Taxa | Aligned embeddings | | | DNA-to-DNA | | | Image-to-Image | | | Image-to-DNA | | |
|---|---|---|---|---|---|---|---|---|---|---|---|---|
| | Img | DNA | Txt | Seen | Unseen | H.M. | Seen | Unseen | H.M. | Seen | Unseen | H.M. |
| Order | ✗ | ✗ | ✗ | 98.6 | 81.5 | 89.2 | 54.5 | 39.7 | 45.9 | 8.4 | 6.0 | 7.0 |
| | ✓* | ✗ | ✗ | — | — | — | 59.6 | 55.4 | 57.4 | — | — | — |
| | ✓ | ✗ | ✓ | — | — | — | 89.2 | 85.9 | 87.5 | — | — | — |
| | ✓ | ✓ | ✗ | **100.0** | **100.0** | **100.0** | **99.5** | *94.1* | *96.7* | **99.5** | *72.0* | *83.5* |
| | ✓ | ✓ | ✓ | **100.0** | **100.0** | **100.0** | 98.6 | **96.1** | **97.3** | 99.2 | **76.0** | **86.1** |
| Family | ✗ | ✗ | ✗ | 87.0 | 75.8 | 81.0 | 29.3 | 23.4 | 26.0 | 0.5 | 0.5 | 0.5 |
| | ✓* | ✗ | ✗ | — | — | — | 39.9 | 31.0 | 34.9 | — | — | — |
| | ✓ | ✗ | ✓ | — | — | — | *90.1* | 74.7 | 81.7 | — | — | — |
| | ✓ | ✓ | ✗ | 99.9 | *96.4* | *98.1* | 89.6 | *78.6* | *83.7* | **92.2** | *48.5* | *63.6* |
| | ✓ | ✓ | ✓ | **100.0** | **97.9** | **98.9** | **92.9** | **79.7** | **85.8** | 88.6 | **54.5** | **67.5** |
| Genus | ✗ | ✗ | ✗ | 81.2 | 67.4 | 73.7 | 13.8 | 11.4 | 12.5 | 0.1 | 0.0 | 0.0 |
| | ✓* | ✗ | ✗ | — | — | — | 23.3 | 17.3 | 19.9 | — | | |
| | ✓ | ✗ | ✓ | — | — | — | 69.7 | 53.1 | 60.3 | — | — | — |
| | ✓ | ✓ | ✗ | *98.1* | *93.1* | *95.5* | 75.4 | *61.7* | *67.9* | **73.2** | *23.3* | *35.3* |
| | ✓ | ✓ | ✓ | **99.0** | **95.7** | **97.3** | **76.0** | **63.1** | **69.0** | 68.6 | **25.5** | **37.2** |
| Species | ✗ | ✗ | ✗ | 76.4 | 62.2 | 68.6 | 7.8 | 5.3 | 6.3 | 0.0 | 0.0 | 0.0 |
| | ✓* | ✗ | ✗ | — | — | — | 13.7 | 9.5 | 11.2 | — | — | — |
| | ✓ | ✗ | ✓ | — | — | — | 52.4 | 36.9 | 43.3 | — | — | — |
| | ✓ | ✓ | ✗ | *95.8* | *87.3* | *91.4* | **61.9** | *46.0* | **52.8** | **59.3** | *9.6* | *16.5* |
| | ✓ | ✓ | ✓ | **97.1** | **90.2** | **93.5** | *60.2* | **46.5** | 52.5 | 52.1 | **10.3** | **17.2** |

Table 10: Top-1 *micro* accuracy (%) on our *val* set for BIOSCAN-1M and different combinations of aligned embeddings (image, DNA, text) during contrastive training.

| Taxa | Aligned embeddings | | | DNA-to-DNA | | | Image-to-Image | | | Image-to-DNA | | |
|---|---|---|---|---|---|---|---|---|---|---|---|---|
| | Img | DNA | Txt | Seen | Unseen | H.M. | Seen | Unseen | H.M. | Seen | Unseen | H.M. |
| Order | ✗ | ✗ | ✗ | 99.2 | 98.4 | 98.8 | 89.3 | 90.7 | 90.0 | 32.2 | 29.8 | 31.0 |
| | ✓* | ✗ | ✗ | — | — | — | 94.2 | 94.3 | 94.2 | — | — | — |
| | ✓ | ✗ | ✓ | — | — | — | **99.7** | **99.6** | **99.6** | — | — | — |
| | ✓ | ✓ | ✗ | **100.0** | **100.0** | **100.0** | **99.7** | **99.6** | **99.6** | *99.6* | *98.9* | *99.2* |
| | ✓ | ✓ | ✓ | **100.0** | **100.0** | **100.0** | **99.7** | *99.5* | **99.6** | **99.7** | **99.0** | **99.3** |
| Family | ✗ | ✗ | ✗ | 96.4 | 94.2 | 95.3 | 54.6 | 61.7 | 57.9 | 2.9 | 3.7 | 3.3 |
| | ✓* | ✗ | ✗ | — | — | — | 67.6 | 71.9 | 69.7 | — | — | — |
| | ✓ | ✗ | ✓ | — | — | — | 95.9 | 92.9 | 94.4 | — | — | — |
| | ✓ | ✓ | ✗ | *99.8* | *99.4* | *99.6* | 95.9 | *93.3* | *94.6* | *95.9* | *85.7* | *90.5* |
| | ✓ | ✓ | ✓ | **100.0** | **99.7** | **99.8** | **96.5** | **94.3** | **95.4** | **96.5** | **86.8** | **91.4** |
| Genus | ✗ | ✗ | ✗ | 92.9 | 89.0 | 90.9 | 30.3 | 41.4 | 35.0 | 0.4 | 0.3 | 0.3 |
| | ✓* | ✗ | ✗ | — | — | — | 42.9 | 51.4 | 46.8 | — | — | — |
| | ✓ | ✗ | ✓ | — | — | — | 87.1 | 79.3 | 83.0 | — | — | — |
| | ✓ | ✓ | ✗ | *99.2* | *97.2* | *98.2* | 88.9 | *83.6* | *86.2* | **87.2** | *58.2* | *69.8* |
| | ✓ | ✓ | ✓ | **99.5** | **98.2** | **98.8** | **89.6** | **84.5** | **87.0** | *86.4* | **59.8** | **70.7** |
| Species | ✗ | ✗ | ✗ | 89.5 | 84.8 | 87.1 | 18.1 | 31.6 | 23.0 | 0.1 | 0.1 | 0.1 |
| | ✓* | ✗ | ✗ | — | — | — | 28.4 | 41.1 | 33.6 | — | — | — |
| | ✓ | ✗ | ✓ | — | — | — | 76.1 | 68.0 | 71.8 | — | — | — |
| | ✓ | ✓ | ✗ | *98.1* | *95.2* | *96.6* | *79.7* | *74.0* | *76.7* | **75.4** | *38.8* | *51.2* |
| | ✓ | ✓ | ✓ | **98.8** | **96.5** | **97.6** | **80.0** | **74.3** | **77.0** | *73.3* | **39.6** | **51.4** |

**Results for top-1 micro-accuracy and validation set.** For completeness, we provide the top 1 micro-accuracy on the test set (Table 8), and results on the validation set (see Table 9 for macro accuracy, and Table 10 for micro accuracy).

Overall, we see a similar trend in results as for macro accuracy on the test set (see Table 1 in the main paper), with the trimodal model that aligns image (I), DNA (D), and text (T) performing the best, and the I+D model outperforming the I+T model. We also observe that the micro averages (over individual samples) are much higher than the macro averages (over classes). This is expected as the rare classes are more challenging and pulls down the macro-average.

We further examine the reproducibility of our results across training runs. In Figure 9, we plot the mean and standard deviation of the micro and macro accuracy evaluated on our BIOSCAN-1M validation set after training with four different random seeds. The results indicate that although there is some fluctuation, the results are mostly stable across training runs. As expected, we observe higher variation for macro accuracy as there are many classes with only a few query records (see Figure 8). There is also more variation for image-to-DNA, where the embeddings are less aligned.

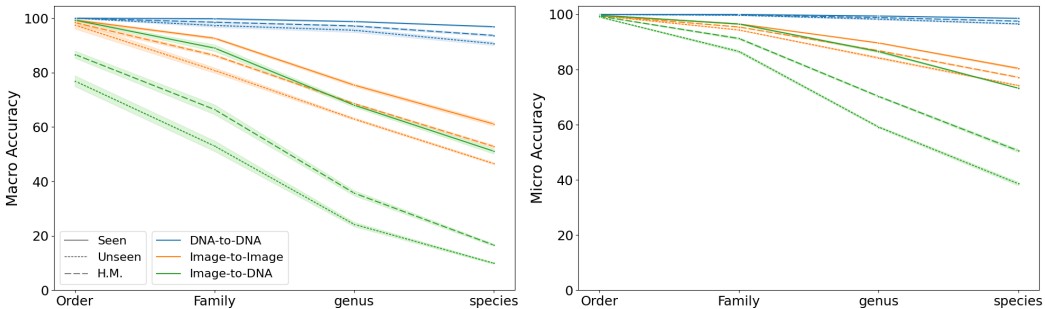

Figure 9: Performance across four training runs for our I+D+T model that aligns all three embeddings during contrastive training. We plot the mean with the shaded regions for standard error for top-1 macro and micro accuracy (%) on our *val* set for BIOSCAN-1M. These plots clearly show that accuracy drops as we go from order to species, and that DNA-to-DNA retrieval is the most accurate with image-to-image second. Overall, the standard error is within 0.5% for the micro accuracy, with higher variation in the image-to-DNA setting.

**Unimodal contrastive loss.** To check whether the improved unimodal performance is a result of cross-modal contrastive learning between DNA and image, or just an artifact of training on the domain-specific data, we conduct experiments comparing our results with fine-tuning the image encoder with unimodal contrastive learning. We use SimCLR-style [11] training on the image encoder with normalized temperature scaled cross-entropy loss (NT-XEnt), and form positive pairs by using data augmentation. For data augmentation, we used the same set of augmentations as [11] by following SimCLR's code, which focused on enhancing image variability. The augmentation process begins with randomly cropping the image to a size of 224, and includes random horizontal flips and random applications of color jitter with an 80% probability. Additionally, images are converted to grayscale 20% of the time, and Gaussian blur is applied with a kernel size equal to 10% of the image size, which is 22 in our case.

We see from the results (denoted by ✓* in Tables 1, 8, 9, and 10), that while SimCLR training improves performance for image-to-image accuracy, it greatly underperforms our CLIBD multimodal model that aligns image to DNA. This shows that while training on the BIOSCAN-1M dataset helps to improve the otherwise generic image encoder, the signal from other modalities has the greatest benefit to performance.

### B.1.1 DIFFERENT TRAINING STRATEGIES

**CLIBD with SimCLR pretraining.** We investigate whether starting with an image encoder that is already pretrained on BIOSCAN-1M images can improve the performance of our CLIBD model. In Table 11, we show that starting with a image encoder that is fine-tuned on BIOSCAN-1M using SimCLR, and then performing full CLIBD training (by aligning image, DNA, and taxonomic labels), we achieve improved image-to-image performance, and image-to-DNA performance at finer taxonomic levels.

**ImageBind-style alignment.** To determine whether it is necessary to align all the modalities to each other, or whether it would be better use a ImageBind-style [25] training where we freeze the image and text encoders and align the DNA encoder. For these experiments, we use BioCLIP's image and text encoder (which are pre-aligned) and BarcodeBERT as DNA encoder.

Table 11: Top-1 *macro* accuracy (%) on the *test* set of BIOSCAN-1M comparing starting with pre-trained image encoder that was fine-tuned using SimCLR (✓)or not (✗). We do full CLIBD training that align all three modalities (the I+D+T model).

| Taxa | SimCLR | DNA-to-DNA | | | Image-to-Image | | | Image-to-DNA | | |
|---|---|---|---|---|---|---|---|---|---|---|
| | | Seen | Unseen | H.M. | Seen | Unseen | H.M. | Seen | Unseen | H.M. |
| Order | ✗ | **100.0** | **100.0** | **100.0** | **99.7** | 94.4 | 97.0 | 99.4 | **88.5** | **93.6** |
| | ✓ | **100.0** | **100.0** | **100.0** | **99.7** | 98.9 | 99.3 | 99.6 | 75.3 | 84.8 |
| Family | ✗ | **100.0** | 98.3 | 99.1 | 90.9 | 81.8 | 86.1 | 90.8 | 50.1 | 64.6 |
| | ✓ | **100.0** | 99.5 | 99.7 | 92.2 | 85.8 | 88.9 | 92.6 | 52.1 | 66.7 |
| Genus | ✗ | 98.2 | **94.7** | **96.4** | 74.6 | 60.4 | 66.8 | 70.6 | 20.8 | 32.1 |
| | ✓ | 98.5 | 94.5 | **96.4** | 77.2 | 63.5 | 69.6 | 73.5 | 23.4 | 35.5 |
| Species | ✗ | 95.6 | **90.4** | **92.9** | 59.3 | 45.0 | 51.2 | 51.6 | 8.6 | 14.7 |
| | ✓ | **95.7** | 89.8 | 92.7 | 62.2 | 47.9 | 54.1 | 56.5 | 9.7 | 16.5 |

Table 12: Top-1 *macro* accuracy (%) on the *test* set of BIOSCAN-1M, binding DNA to frozen (❄) image and text encoders. We compare binding only to the image (following ImageBind [25]) or aligning to both image and text. Here, we utilized BioCLIP's image and text encoders, with BarcodeBERT for DNA encoding.

| Taxa | Bind to | Frozen | | | DNA-to-DNA | | | Image-to-Image | | | Image-to-DNA | | |
|---|---|---|---|---|---|---|---|---|---|---|---|---|---|
| | | Img | DNA | Txt | Seen | Unseen | H.M. | Seen | Unseen | H.M. | Seen | Unseen | H.M. |
| Order | Image | ❄ | ✗ | ❄ | 100.0 | 100.0 | 100.0 | 88.5 | 86.0 | 87.2 | 87.8 | 64.8 | 74.6 |
| | Image & Text | ❄ | ✗ | ❄ | 100.0 | 100.0 | 100.0 | 88.5 | 86.0 | 87.2 | **88.7** | **71.9** | **79.4** |
| Family | Image | ❄ | ✗ | ❄ | 100.0 | 97.6 | 98.8 | 84.3 | 68.4 | 75.5 | **79.6** | 43.1 | 55.9 |
| | Image & Text | ❄ | ✗ | ❄ | 100.0 | 97.2 | 98.6 | 84.3 | 68.4 | 75.5 | 78.1 | **45.4** | **57.5** |
| Genus | Image | ❄ | ✗ | ❄ | 98.2 | 92.9 | 95.5 | 62.6 | 47.1 | 53.7 | 56.0 | 17.9 | 27.1 |
| | Image & Text | ❄ | ✗ | ❄ | 98.9 | 93.2 | 96.0 | 62.6 | 47.1 | 53.7 | **56.5** | **19.7** | **29.2** |
| Species | Image | ❄ | ✗ | ❄ | 95.1 | **87.9** | 91.3 | 44.1 | 32.4 | 37.4 | **39.5** | 7.4 | 12.5 |
| | Image & Text | ❄ | ✗ | ❄ | **95.9** | 88.6 | **92.1** | 44.1 | 32.4 | 37.4 | 38.0 | **8.5** | **13.9** |

The first approach follows the ImageBind paradigm, where the DNA encoder (BarcodeBERT) is aligned only to BioCLIP's image encoder. The second approach follows our CLIBD protocol and allows the unfrozen DNA encoder to align simultaneously with BioCLIP's image encoder and text encoder. The results show that allowing the DNA encoder to align to both the image and text encoders vs. just binding to the image encoder results in slightly higher performance. Compared to the results in Table 11, it is evident that these approaches underperform compared to our proposed method CLIBD across all query and key combinations. Note that since we start with BioCLIP's pretrained models, and since BioCLIP's training data includes BIOSCAN-1M, it is not guaranteed that *unseen* species labels are actually unseen during training.

### B.1.2 COMPARISON WITH BIOCLIP

**Full taxonomic comparison with BioCLIP.** In Table 13, we compare the performance of BioCLIP vs. CLIBD on classification for all four different taxonomic levels. We see that our CLIBD with I+T is able to outperform BioCLIP at every taxonomic level. From Table 1, we see that the I+D model can already outperform our I+T model. By incorporating DNA barcodes (I+D+T), we are able to achieve the best performance.

### B.1.3 USING LARGER MODELS

**Experiments with OpenCLIP.** We conduct experiments using OpenCLIP as our text and image encoder, as well as larger ViT and BERT models. We train our full trimodal model (with image, DNA, text alignment), and report the species-level top-1 macro accuracy on our validation set for BIOSCAN-1M in Table 14.

Table 13: Comparison of top-1 macro-accuracy (%) of BioCLIP and our CLIBD model on the test set, matching image embeddings (queries) against embeddings of different modalities for retrieval (image, DNA, and text keys). *Note:* the BioCLIP model [61] was trained on data that included BIOSCAN-1M but used different species splits, so it may have seen most of the unseen species during its training.

| Taxon. level | Model | Aligned embeddings | | | Image-to-Image | | | Image-to-DNA | | | Image-to-Text | | |
|---|---|---|---|---|---|---|---|---|---|---|---|---|---|
| | | Img | DNA | Txt | Seen | Unseen | H.M. | Seen | Unseen | H.M. | Seen | Unseen | H.M. |
| Order | BioCLIP | ✓ | ✗ | ✓ | 73.3 | 69.2 | 71.2 | — | — | — | 38.6 | 35.5 | 37.0 |
| | CLIBD | ✓ | ✗ | ✓ | *99.6* | *97.4* | *98.5* | — | — | — | **99.6** | **77.4** | **87.1** |
| | CLIBD | ✓ | ✓ | ✓ | **99.7** | *94.4* | *97.0* | 99.4 | 88.5 | 93.6 | 99.2 | 74.7 | 85.2 |
| Family | BioCLIP | ✓ | ✗ | ✓ | 56.9 | 42.2 | 48.5 | — | — | — | 18.9 | 14.6 | 16.4 |
| | CLIBD | ✓ | ✗ | ✓ | *90.7* | *76.7* | *83.1* | — | — | — | *94.9* | *49.9* | *65.4* |
| | CLIBD | ✓ | ✓ | ✓ | **90.9** | **81.8** | **86.1** | 90.8 | 50.1 | 64.6 | **95.8** | **50.9** | **66.5** |
| Genus | BioCLIP | ✓ | ✗ | ✓ | 33.8 | 24.3 | 28.3 | — | — | — | 9.5 | 5.9 | 7.3 |
| | CLIBD | ✓ | ✗ | ✓ | *72.1* | *49.6* | *58.8* | — | — | — | *81.9* | **81.9** | *29.7* |
| | CLIBD | ✓ | ✓ | ✓ | **74.6** | **60.4** | **66.8** | 70.6 | 20.8 | 32.1 | **83.0** | *21.6* | **34.3** |
| Species | BioCLIP | ✓ | ✗ | ✓ | 20.4 | 14.8 | 17.1 | — | — | — | 4.2 | 3.1 | 3.6 |
| | CLIBD | ✓ | ✗ | ✓ | *54.2* | *33.6* | *41.5* | — | — | — | **57.6** | *4.6* | *8.5* |
| | CLIBD | ✓ | ✓ | ✓ | **59.3** | **45.0** | **51.2** | 51.6 | 8.6 | 14.7 | *56.0* | *4.8* | **8.9** |

Table 14: Species-level top-1 macro accuracy (%) on our *val* set for BIOSCAN-1M with CLIBD using different image and text encoders. Results use image embedding to match against different embeddings for retrieval (Image, DNA, and Text). We compare using the OpenCLIP (OC) pretrained model with other models. For these experiments, we used OpenCLIP ViT-L/14 [30] which is pre-trained on OpenAI's dataset that combines multiple pre-existing image datasets such as YFCC100M [66]. For timm ViT-B/16 (`vit_base_patch16_224`) and timm ViT-L/16 (`vit_large_patch16_224`) [73] both are trained on ImageNet [17]. We also used `bert-base-uncased` as our text encoder, which was pretrained on BookCorpus [81]. For the DNA encoder, we use BarcodeBERT (except for the first row, where we do not align the DNA embeddings). We highlight in gray the setting that uses the same vision and text encoder that we used in our other experiments.

| OC | Batch size | Epoch | Training time (per epoch) | Memory CUDA | Aligned embeddings | | | Image-to-Image | | | Image-to-DNA | | | Image-to-Text | | |
|---|---|---|---|---|---|---|---|---|---|---|---|---|---|---|---|---|
| | | | | | Image | DNA | Text | Seen | Unseen | H.M. | Seen | Unseen | H.M. | Seen | Unseen | H.M. |
| ✓ | 200 | 15 | 1.3 hour | 70.1GB | OpenCLIP(L/14) | ✗ | OpenCLIP | 54.4 | 36.7 | 43.8 | — | — | — | **53.9** | *7.1* | *12.6* |
| ✓ | 200 | 15 | 1.4 hour | 84.1GB | OpenCLIP(L/14) | ✓ | OpenCLIP | 56.8 | **41.1** | **47.7** | 36.4 | 9.0 | 14.4 | 51.2 | **7.5** | **13.0** |
| ✗ | 200 | 15 | 1.5 hour | 37.4GB | timm(B/16) | ✓ | BERT(small) | 52.7 | 37.7 | 44.0 | 32.1 | 7.0 | 11.5 | 44.7 | 5.2 | 9.4 |
| ✗ | 500 | 38 | 0.6 hour | 82.1GB | timm(B/16) | ✓ | BERT(small) | **57.8** | 40.2 | *47.5* | **44.5** | **9.8** | **16.0** | *51.4* | 6.1 | 10.9 |
| ✗ | 200 | 15 | 1.5 hour | 72.5GB | timm(L/16) | ✓ | BERT(base-uncased) | 55.9 | 40.1 | 46.7 | 34.5 | 8.1 | 13.1 | 41.1 | 6.3 | 10.9 |

We select OpenCLIP ViT-L/14 [30] as a representative of a pretrained vision-language model that is trained with contrastive loss. As the OpenCLIP model requires a large amount of memory, we use a batch size of 200. From Table 14, we see that using OpenCLIP (first two rows), we do achieve better performance (especially for image to text) compared to our choice of Timm VIT B/16 and BERT-small for the image and text encoder at batch size of 200 (row 3). To disentangle whether the better performance is from the prealigned image and text embeddings or from the larger model size, we compare with training with a larger batch size (with similar CUDA memory usage, row 4) and larger unaligned image and text encoder, e.g. Timm ViT-L/16 and Bert-Base (row 5). Using a larger batch size brings the image-to-image performance close to that of the model with OpenCLIP (row 2), and can be improved even further with larger batch size (see Table 20). However, the image-to-text performance is still lower, indicating that the pretrained aligned image-to-text model is helpful despite the domain gap between the taxonomic labels and the text that makes up most of the pretraining data for OpenCLIP.

## B.2 ADDITIONAL CROSS-MODAL RETRIEVAL RESULTS

**Details about the seen/unseen classifier for the IS-DU strategy.** For the NN classifier, we compute the cosine similarity of the image query features with the image features of the seen species. If the most similar image key has a similarity higher than threshold $t_1$, it is considered *seen*. In the supervised fine-tuning approach, we add a linear classifier after the image encoder and fine-tune the encoder and classifier to predict the species out of the set of seen species. If the softmax probability exceeds $t_2$, the image is classified as *seen*.

We tuned $t_1$ and $t_2$ on the validation set using a uniform search over 1000 values between 0 and 1, maximizing the harmonic mean of the accuracy for seen and unseen species. We report the binary

Table 15: Accuracy (%) of our I+D+T model in predicting whether an image query corresponds to a seen or unseen species, as a binary classification problem (evaluated on our BIOSCAN-1M test set). For the "DNA" strategy with nearest neighbour (NN), we use the nearest DNA feature to classify into seen or unseen. It serves as a form of "oracle" as it has access to the samples from unseen species. For the "IS+DU" strategy and NN, we threshold the highest cosine similarity score against image keys. For the supervised linear classifier (Linear), we threshold the confidence score of the prediction over seen species. We report accuracy for seen and unseen species, and their harmonic mean (H.M).

| Method | Strategy | Seen | Unseen | H.M. |
|---|---|---|---|---|
| NN (oracle) | DNA | 82.16 | 76.21 | 79.07 |
| NN | IS+DU | **83.29** | 76.83 | **79.93** |
| Linear | IS+DU | 73.27 | **85.14** | 78.76 |

Table 16: Top-1 accuracy (%) on our BIOSCAN-1M test set using the Image+DNA+Text model with image query. We compare nearest neighbour (NN) using only DNA keys, vs. our two strategies to use Image key for seen and DNA key for Unseen, either NN or a supervised linear classifier. We also compare against BZSL [4] with our embeddings.

| Taxa | Method | Strategy | Micro top-1 acc | | | Macro top-1 acc | | |
|---|---|---|---|---|---|---|---|---|
| | | | Seen | Unseen | H.M. | Seen | Unseen | H.M. |
| Order | NN | DNA | **99.7** | **99.3** | **99.5** | **99.4** | 88.5 | 93.6 |
| | NN | IS+DU | 99.4 | 99.2 | 99.3 | 99.3 | **89.3** | **94.1** |
| | Linear | IS+DU | 99.5 | 99.2 | 99.3 | 99.3 | 88.3 | 93.5 |
| | BZSL | IS+DU | 99.4 | 98.2 | 98.8 | 98.9 | 59.6 | 74.4 |
| Family | NN | DNA | **96.5** | **87.1** | **91.6** | **90.8** | 50.1 | **64.6** |
| | NN | IS+DU | 94.6 | **87.1** | 90.7 | 83.0 | **52.6** | 64.4 |
| | Linear | IS+DU | 94.7 | 87.0 | 90.7 | 82.8 | 51.2 | 63.3 |
| | BZSL | IS+DU | 95.6 | 80.3 | 87.3 | 88.0 | 32.5 | 47.5 |

classification results on our BIOSCAN-1M test set in Table 15. In these experiments, we use the I+D+T model with images as the queries.

**BSZL details** In BZSL [4], the authors used a two-layer Bayesian model with features extracted from pretrained image and DNA encoders. The model relies on hyperparameters, which shape the size and scaling of the posterior distribution used in Bayesian inference and thus significantly affect its performance. Tuning the hyperparameters on the validation split of the INSECT dataset allows for a practioner to tradeoff between accuracy for seen vs. unseen classes. In our experiments, we use the same search space as in the original paper for our hyperparameter tuning, which takes about 4 hours. We find that the resulting accuracy is volatile and that expanding the search space can lead to an impractical search time.

**Order and family results for BIOSCAN-1M.** In Table 16, we report the performance of the direct image-to-DNA matching (NN with DNA), as well as our IS+DU strategy (with the NN and linear classifiers), as well as BZSL with embeddings from our CLIBD. In the IS-DU strategy, for both the NN and linear classifier, if a image is classified as *seen*, we will use image-to-image matching to identify the most similar key, and classify the species using that key. Otherwise, we match the image query features with the DNA key features for unseen species.

Results show that at the order and family-level, direct image-to-DNA matching and NN with IS-DU gives the highest performance, with BZSL being the worst performing.

## B.3 Embedding space visualization

To better understand the alignment of features in the embedding space, we visualize a mapping of the image, DNA, and text embeddings in Figure 10. We use UMAP [42] with a cosine similarity metric applied to the seen validation set to map the embeddings down to 2D space, and we mark points in the space based on their order classification. We show the embedding space before alignment (a),

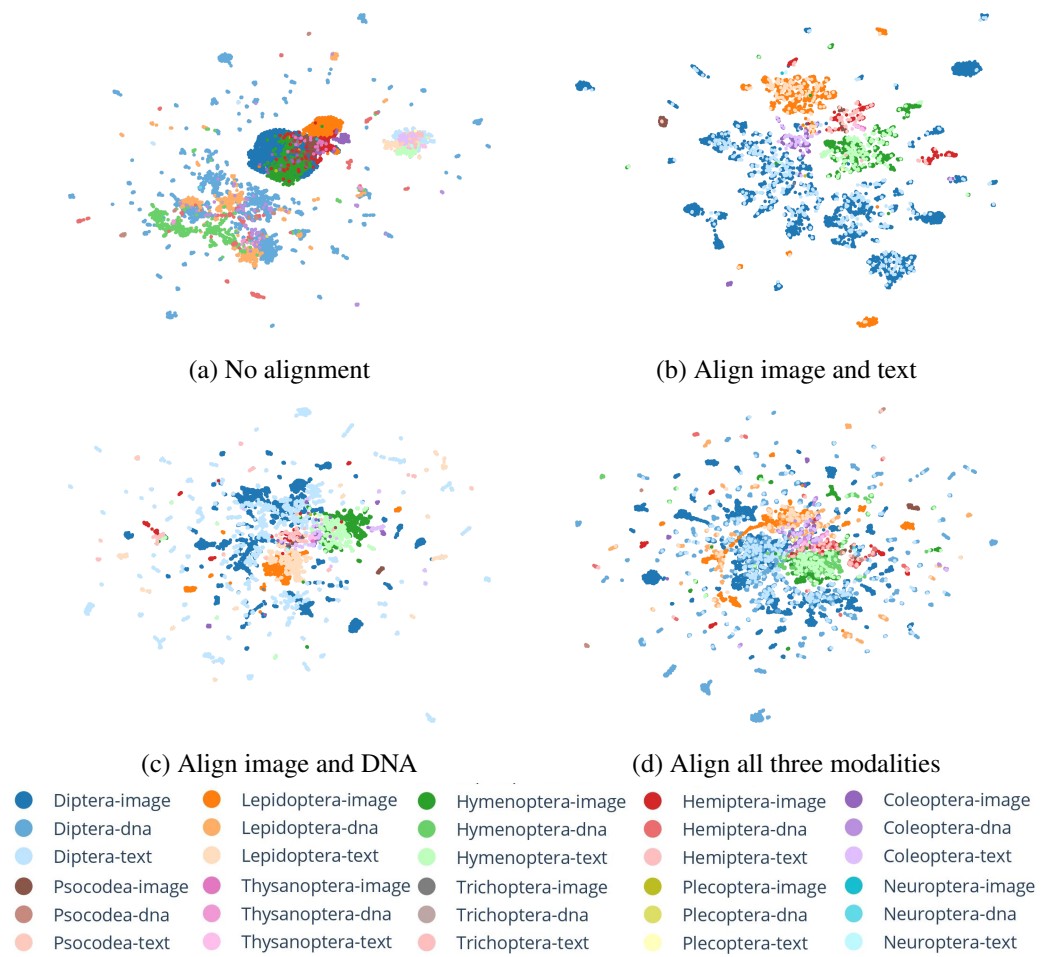

(a) No alignment

(b) Align image and text

(c) Align image and DNA

(d) Align all three modalities

| | | |
|---|---|---|
| ● Diptera-image | ● Lepidoptera-image | ● Hymenoptera-image |
| ● Diptera-dna | ● Lepidoptera-dna | ● Hymenoptera-dna |
| ● Diptera-text | ● Lepidoptera-text | ● Hymenoptera-text |
| ● Psocodea-image | ● Thysanoptera-image | ● Trichoptera-image |
| ● Psocodea-dna | ● Thysanoptera-dna | ● Trichoptera-dna |
| ● Psocodea-text | ● Thysanoptera-text | ● Trichoptera-text |

● Hemiptera-image     ● Coleoptera-image
● Hemiptera-dna       ● Coleoptera-dna
● Hemiptera-text      ● Coleoptera-text
● Plecoptera-image    ● Neuroptera-image
● Plecoptera-dna      ● Neuroptera-dna
● Plecoptera-text     ● Neuroptera-text

Figure 10: *Embedding visualization for order level.* We visualize the embedding space with **no alignment** (a), **image and text** aligned (b), **image and DNA** aligned (c), and **all three modalities** aligned (d) over the seen validation set generated using UMAP on the image, DNA, and text embeddings, using a cosine similarity distance metric. Marker hue: order taxon. Marker lightness: data modality.

with image and text (b), image and DNA (c), and all three modalities. We see that after aligning the modalities, samples for the same order (indicated by hue), from different modalities (indicated by lightness) tend to overlap each other. We observe that, for some orders, there are numerous outlier clusters spread out in the space. However, overall the orders demonstrate some degree of clustering together, with image and DNA features close to one another within their respective clusters. Furthermore, we note the text embeddings tend to lie within the Image or (more often) DNA clusters, suggesting a good alignment between text and other modalities.

### B.4 ATTENTION MAP VISUALIZATION

We provide more attention map visualization samples in Figure 11, Figure 12 and Figure 13, including both success cases and failure cases. Beneath each attention map, we show the closest retrieval for that model. From the figure, we see that the retrieved images are all highly similar to the query image. This illustrates the challenges of using image alone to perform taxonomic classification.

## C   IMPLEMENTATION DETAILS AND HYPERPARAMETER SELECTION

In this section, we provide experiments to validate the choice of hyperparameter settings and design choices we made for efficient training of our model.

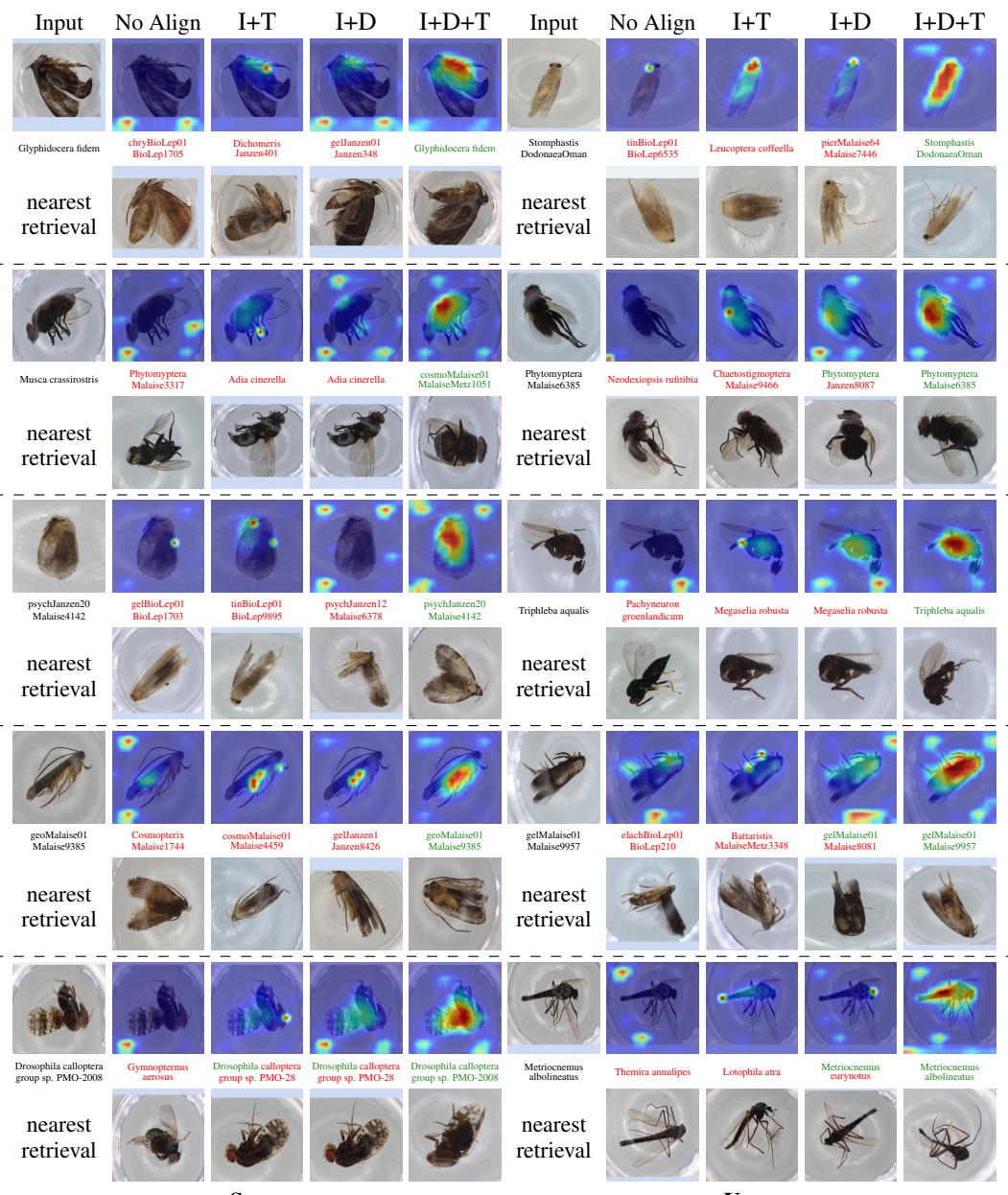

Figure 11: We visualize the attention and the nearest retrievals for queries from seen and unseen species. Predicted genera + species are indicated green text for correct, red for incorrect. Here, we present the cases where only the I+D+T (Image+DNA+Text) model gives the correct results. In these cases, the attention map of the I+D+T model's image encoder activates well on the model's prediction results, while the other models largely do not attend visually to the entire insect.

## C.1 TRAINABLE VS FIXED TEMPERATURE

We compare using a fixed temperature for the contrastive loss vs using trainable temperature (Table 17). We find that using the trainable temperature helps improve the performance, provided the model is trained for enough epochs.

## C.2 AUTOMATIC MIXED PRECISION

For efficient training with large batch sizes, we use automatic mixed precision (AMP) with the bfloat16 data type. The bfloat16 data type gives a similar dynamic range as float32 at reduced precision, and provides stable training with reduced memory usage.

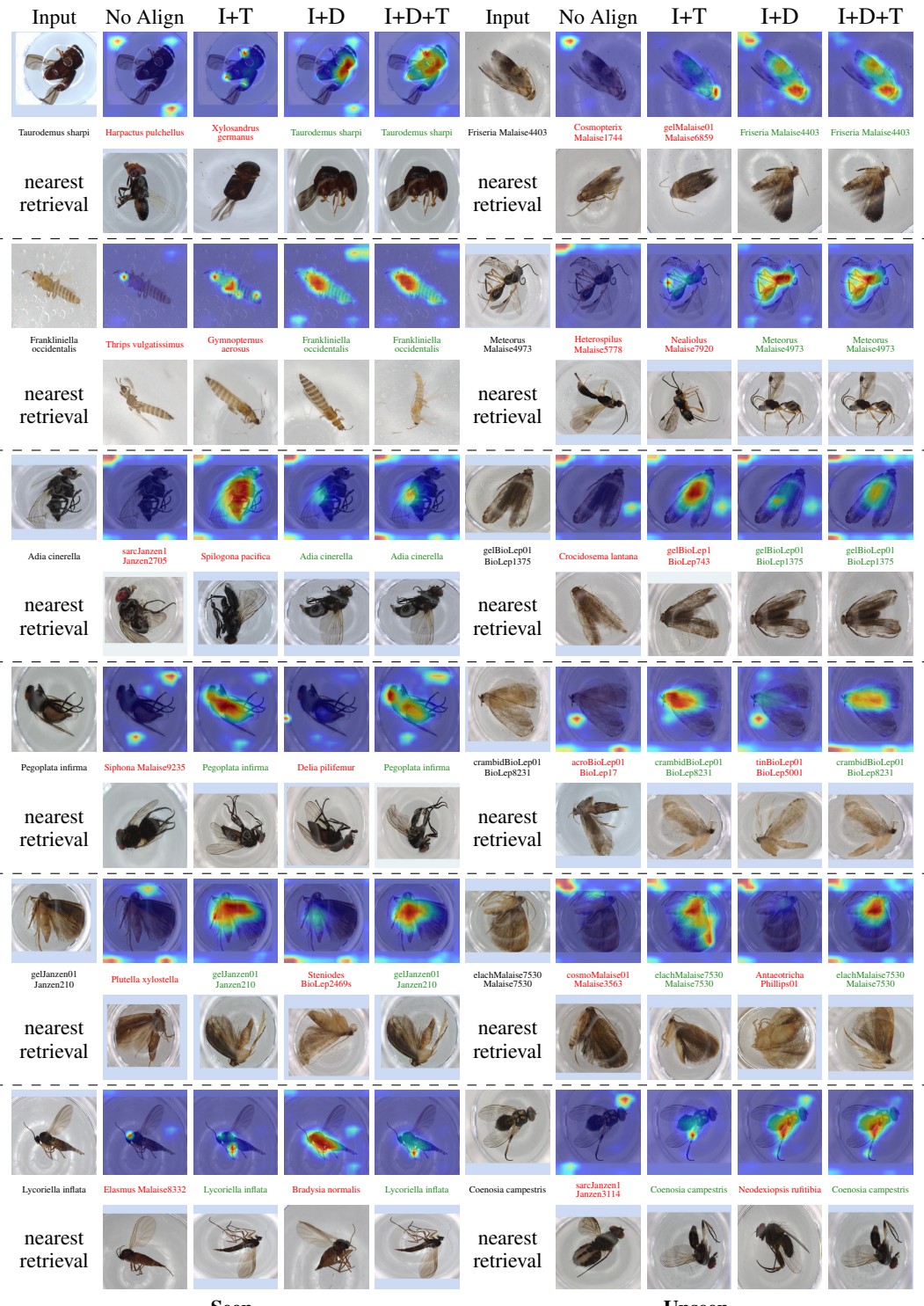

Figure 12: We visualize the attention and the nearest retrievals for queries from seen and unseen species for which multiple cross-modally aligned models correctly predicted the species (top three are examples with I+D correct, bottom three are examples with I+T correct). Predicted genera + species are indicated green text for correct, red for incorrect. For correct predictions, the attention map more often signals on the whole insect, and the nearest retrievals found by the correct models are more reasonable than the incorrect ones. Interestingly, we see that the strong activation correlate with incorrect predictions sometimes (5th,6th examples for incorrect I+T prediction, and last two examples for poor I+D prediction).

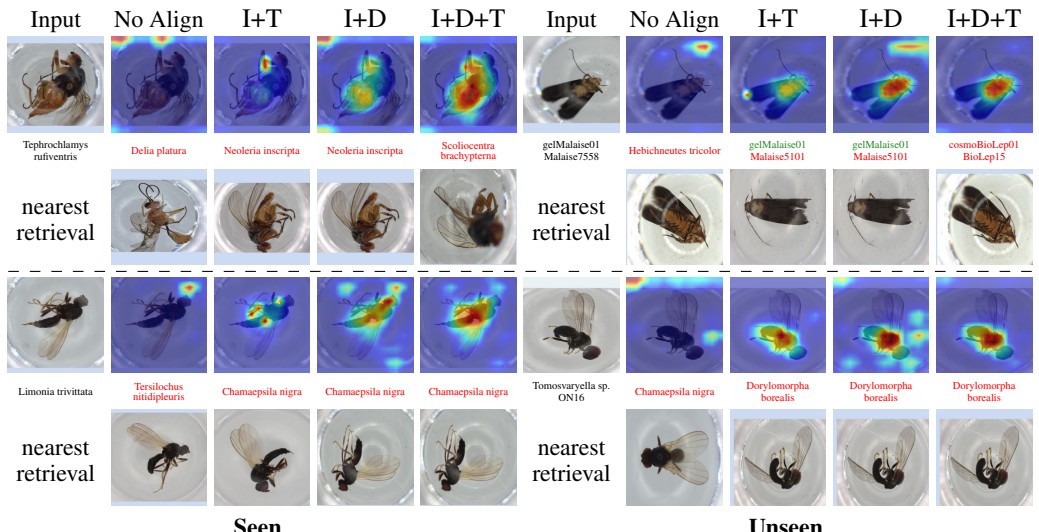

Figure 13: *Failure cases.* We visualize the attention and the nearest retrievals for queries from seen and unseen species for incorrect predictions by all four models (no align, I+T, I+D, I+D+T). Predicted genera + species are indicated in green text for correct, red for incorrect. Despite strong signalling on the insect, the model identifies alternative samples as the nearest, though the retrieved examples still exhibit high-level visual similarities.

Table 17: *Trainable temperature.* Top-1 accuracy on the validation set for models contrastively trained with either a fixed or trainable temperature. We consider the performance when training for different durations (1 and 15 epochs).

| Temperature | Epochs | Micro Top-1 Accuracy (%) | | | Macro Top-1 Accuracy (%) | | |
|---|---|---|---|---|---|---|---|
| | | DNA-to-DNA | Image-to-Image | Image-to-DNA | DNA-to-DNA | Image-to-Image | Image-to-DNA |
| Fixed | 1 | **97.2** | 62.6 | **27.4** | **93.1** | 41.6 | **11.0** |
| Trainable | 1 | 96.8 | **63.5** | 24.2 | 92.0 | **42.8** | 10.4 |
| Fixed | 15 | 98.0 | 74.2 | **57.8** | **94.7** | 52.2 | 32.4 |
| Trainable | 15 | **98.4** | **76.9** | 56.5 | 94.1 | **57.8** | **35.7** |

We compare training with and without AMP in Table 18. By applying AMP, we achieve comparable performance while using less memory. With AMP, the CUDA memory usage is reduced by about 15GB (~20%) and the training time by 3 hour per epoch (~75%). Although using full-precision (no AMP) yields slightly better accuracies, the lower memory usage and faster training time of AMP allows for more efficient experiments. Additionally, the lower memory usage with AMP enables us to use larger batch sizes and is more effective for our experiments.

## C.3 LoRA vs full fine-tuning

For efficient training, we also investigate the performance of using LoRA [28] vs full fine-tuning. As shown in Table 19, we find that while LoRA does reduce the memory usage and training time, the performance is also notably worse, and thus we use full fine-tuning for the rest of our experiments.

Table 18: *Automatic mixed precision.* Top-1 micro and macro accuracy on the validation set with models contrastively trained either with or without automatic mixed precision (AMP). We compare accuracies across different embedding alignments (image-to-image, DNA-to-DNA, and image-to-DNA). Both experiments have otherwise identical training conditions, including a batch size of 300 and 15 training epochs.

| | Micro Top-1 Accuracy (%) | | | Macro Top-1 Accuracy (%) | | | Memory | Training Time |
|---|---|---|---|---|---|---|---|---|
| | DNA-to-DNA | Image-to-Image | Image-to-DNA | DNA-to-DNA | Image-to-Image | Image-to-DNA | CUDA (GB) ↓ | per epoch↓ |
| −AMP | **98.07** | **74.97** | **57.38** | 95.18 | **54.47** | **32.22** | 75.54 | 4.03 hour |
| +AMP | 97.85 | 74.31 | 56.23 | **97.85** | 53.65 | 30.99 | **60.64** | **1.15** hour |

Table 19: *Low-rank adaptation.* Top-1 micro/macro accuracy on the validation set for models contrastively trained with either with full fine-tuning or Low-Rank Adaptation (LoRA). We compare micro and macro Top-1 accuracies across different embedding alignments (image-to-image, DNA-to-DNA, and image-to-DNA). Both strategies use a batch size of 300 and are trained for a total of 15 epochs, allowing us to evaluate the impact of fine-tuning techniques on model performance and CUDA memory usage.

| | Micro Top-1 Accuracy (%) | | | Macro Top-1 Accuracy (%) | | | Memory | Training Time |
|---|---|---|---|---|---|---|---|---|
| Fine-tuning Method | DNA-DNA | Image-Image | Image-DNA | DNA-DNA | Image-Image | Image-DNA | CUDA (GB) ↓ | per epoch↓ |
| Full Fine-Tuning | **98.1** | **74.3** | **58.0** | **95.5** | **54.0** | **32.4** | 78.5GB | 4.03 hour |
| LoRA | 96.2 | 64.9 | 37.6 | 91.3 | 45.4 | 17.1 | **53.4**GB | **2.98** hour |

Table 20: *Batch size.* Top-1 accuracy on the validation set for models contrastively trained with different batch sizes. Training at larger batch sizes helps improve accuracy at more fine-grained taxonomic levels such as genus and species.

| | | Alignment | | | Micro top-1 accuracy | | | | | | | | | Macro top-1 accuracy | | | | | | | | |
| | | | | | DNA to DNA | | | Image to Image | | | Image to DNA | | | DNA to DNA | | | Image to Image | | | Image to DNA | | |
| Taxa | Batch size | Img | DNA | Txt | Seen | Unseen | H.M. | Seen | Unseen | H.M. | Seen | Unseen | H.M. | Seen | Unseen | H.M. | Seen | Unseen | H.M. | Seen | Unseen | H.M. |
|---|---|---|---|---|---|---|---|---|---|---|---|---|---|---|---|---|---|---|---|---|---|---|
| Order | 500 | ✓ | ✓ | ✓ | **100.0** | **100.0** | **100.0** | 99.6 | 99.6 | 99.6 | 99.6 | **99.2** | **99.4** | **100.0** | 92.9 | 96.3 | **99.6** | **98.5** | **99.0** | 99.1 | **75.8** | **85.9** |
| | 1000 | ✓ | ✓ | ✓ | **100.0** | **100.0** | **100.0** | 99.7 | 99.6 | 99.6 | **99.7** | **99.2** | **99.4** | **100.0** | **100.0** | **100.0** | 99.0 | 93.7 | 96.3 | 99.1 | 75.3 | 85.6 |
| | 1500 | ✓ | ✓ | ✓ | **100.0** | **100.0** | **100.0** | 99.7 | 99.6 | **99.7** | **99.7** | **99.2** | **99.4** | **100.0** | 92.8 | 96.3 | 99.5 | 94.3 | 96.8 | **99.6** | 73.6 | 84.6 |
| | 2000 | ✓ | ✓ | ✓ | **100.0** | **100.0** | **100.0** | 99.7 | **99.7** | **99.7** | **99.7** | **99.2** | **99.4** | **100.0** | **100.0** | **100.0** | 99.1 | 95.9 | 97.5 | 99.2 | 73.9 | 84.7 |
| Family | 500 | ✓ | ✓ | ✓ | 99.9 | 99.6 | 99.8 | 95.6 | 93.9 | 94.7 | 94.8 | 86.2 | 90.3 | **100.0** | 97.6 | 98.7 | 88.8 | 79.3 | 83.8 | 83.5 | 52.5 | 64.5 |
| | 1000 | ✓ | ✓ | ✓ | **100.0** | **99.7** | 99.8 | 96.3 | 94.2 | 95.2 | 96.0 | 86.9 | 91.2 | 99.9 | 98.0 | 99.0 | 90.2 | 80.5 | 85.1 | 87.9 | 56.1 | 68.5 |
| | 1500 | ✓ | ✓ | ✓ | **100.0** | 99.6 | 99.8 | 96.5 | 94.3 | 95.4 | **96.7** | 87.3 | **91.8** | **100.0** | 97.3 | 98.6 | **92.0** | 81.3 | **86.3** | **91.7** | 53.9 | 67.9 |
| | 2000 | ✓ | ✓ | ✓ | **100.0** | **99.7** | **99.9** | 96.6 | 94.5 | 95.5 | 96.6 | **87.4** | **91.8** | **100.0** | **98.6** | **99.3** | **92.0** | 81.2 | **86.3** | 90.0 | **56.2** | **69.2** |
| Genus | 500 | ✓ | ✓ | ✓ | 99.2 | **98.3** | 98.8 | 87.5 | 83.6 | 85.5 | 77.3 | 55.9 | 64.9 | 98.4 | 95.5 | 97.0 | 71.2 | 61.6 | 66.1 | 54.0 | 21.4 | 30.7 |
| | 1000 | ✓ | ✓ | ✓ | **99.4** | 97.9 | 98.6 | 88.7 | 84.5 | 86.6 | 82.4 | 58.2 | 68.2 | 98.4 | 94.9 | 96.6 | 74.4 | 63.9 | 68.8 | *61.2* | **24.3** | *34.8* |
| | 1500 | ✓ | ✓ | ✓ | 99.3 | 98.2 | 98.8 | **89.6** | 84.8 | 87.1 | 83.8 | 59.7 | 69.7 | 98.0 | 95.2 | 96.6 | 75.8 | 63.7 | 69.2 | **64.9** | 23.6 | 34.6 |
| | 2000 | ✓ | ✓ | ✓ | **99.4** | **98.4** | **98.9** | **89.6** | **84.9** | **87.2** | **84.8** | **60.1** | **70.3** | **98.9** | **96.5** | **97.7** | **76.1** | **64.2** | **69.6** | **64.9** | *24.0* | **35.0** |
| Species | 500 | ✓ | ✓ | ✓ | 97.9 | **96.5** | 97.2 | 76.8 | 73.7 | 75.2 | 58.8 | 35.8 | 44.5 | 95.5 | 90.7 | 93.0 | 56.4 | 45.9 | 50.6 | 36.5 | 8.7 | 14.0 |
| | 1000 | ✓ | ✓ | ✓ | 98.2 | 96.0 | 97.1 | 78.8 | 74.8 | 76.7 | 67.2 | 37.7 | 48.3 | 95.8 | 89.4 | 92.5 | 59.8 | 47.3 | 52.8 | *44.3* | 9.2 | 15.3 |
| | 1500 | ✓ | ✓ | ✓ | 98.0 | 96.2 | 97.1 | **80.5** | 74.7 | **77.5** | 69.8 | 39.9 | 50.8 | 95.1 | 89.4 | 92.2 | *61.8* | 47.9 | **54.0** | **47.7** | **10.1** | **16.6** |
| | 2000 | ✓ | ✓ | ✓ | **98.5** | **96.5** | **97.5** | 80.0 | **74.9** | *77.3* | **70.5** | **41.3** | **52.1** | **96.9** | **90.9** | **93.8** | *61.3* | **48.0** | *53.9* | **47.7** | *9.9* | *16.4* |

## C.4 BATCH SIZE EXPERIMENTS

We conducted additional experiments to investigate the impact of training batch size (from 500 to 2000) on model performance. The choice of batch size ordinarily does not have a major impact on performance when using supervised learning, but can have larger impact when training using contrastive learning since each positive pair is normalized against the pool of negative pairs appearing in the same training batch.

Our results, shown in Table 20, confirm that the classification accuracy improves as the batch size increases. The effect on is more pronounced for the harder, more fine-grained, taxonomic levels. Due to resource limitations, we were only able to train up to a batch size of 2000. We anticipate that using larger batch sizes would further enhance the classification accuracy of CLIBD, especially on more fine-grained taxonomic levels.

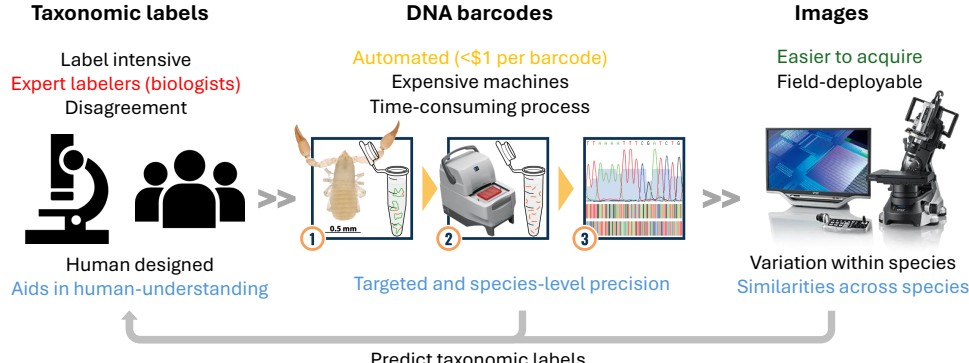

Figure 14: *Overview of modalities.* (a) *Taxonomic labels* rely on expert biologists for annotation, providing valuable insights for human understanding. However, they may be difficult to determine at the species level, especially given bias or expert disagreement. (b) *DNA barcodes* are effective for species identification but require specialized equipment. (c) *Images* are the easiest to acquire, but visual differences between species can be hard to identify. Despite this, they are useful for analyzing similarities and differences between species. We attempt to leverage the benefits of all three modalities to learn a representation space for taxonomic classification and other downstream tasks.

# D   DISCUSSION OF UTILITY AND COST OF DNA BARCODES

In this work, we focus on experiments on BIOSCAN-1M that showcase the value of using DNA barcodes as an alignment target. Here we discuss some limitations of using DNA barcodes as well as their advantages. We also provide details on the acquisition cost, as well as the computational and storage cost of using DNA barcodes.

## D.1   WHEN ARE DNA BARCODES USEFUL?

While DNA barcodes are a promising alignment target, there are also challenges with using them. For one, repositories [52, 6] and datasets [23, 24] that include DNA barcodes are relatively limited. In contrast, there are far more datasets, at larger scale, that consists only of images and taxonomic labels. For instance, Tree-Of-Life10M [61] has more than 10M images of over 454k species and BioTrove [76] has more than 162M images of over 367k species, which is much larger than the recently introduced BIOSCAN-1M and BIOSCAN-5M. In addition, the BIOSCAN datasets are limited to images collected in a lab setting and thus models trained on such data may not readily transfer to images captured in the wild. DNA barcodes are also not perfect indicators of species, especially in cases of complex inheritance (e.g. hybridization). There may also be potential errors during the data collection and barcoding process (e.g. errors during sequencing, potential contamination from other specimens, etc). Thus, relying solely on DNA barcodes for species identification, especially those based purely on fixed thresholds of genetic distances, may lead to errors [43].

Nevertheless, we believe that DNA barcodes can be a useful signal, as validated by our experiments. Since their introduction [26], they have become widely used in large-scale biodiversity monitoring programs as they allow for relatively fast and reliable identification. The use of DNA barcodes for fine-grained species identification is highly accurate, with most errors coming from human errors [12]. With the use of metabarcoding [56], it is also possible to identify multiple species from samples from the environment, allowing for the discovery of new species and monitoring of changes in biodiversity. There are ongoing efforts to scale up the collection of DNA barcoding, as well as developments in technology, such as nanopore sequencing [71], that allows for real-time barcoding of organisms in the field. With these technological advances, we foresee that there will be increasing amount of DNA barcode data in the coming years. These will make DNA barcodes an important modality to leverage for building machine learning models for biodiversity monitoring and species discovery [46, 9, 38]. Outside of biodiversity monitoring, the use of DNA barcodes for identification can be used for dietary analyses (fecal sampling to determine gut content) [59, 49], food authentication (e.g. whether fish is marketed under a different name, use of horse meat, etc.) [20, 27], herbal medicine ingredient identification [10], whether there is illegal transfer of animal remains across borders, and many others.

DNA barcodes can also be especially useful for distinguishing between cryptic species, which are morphologically similar but genetically distinct, and understanding undiscovered species. For specimens that are not yet well established in the standard Linnaean taxonomy, clustering by DNA barcodes can serve as an way to identify a potential related group of organisms, called an Operational Taxonomic Unit. This is standardized through the use of Barcode Index Numbers (BINs) [52].

In our experiments, we attempted to simulate evaluation on *unseen* species by splitting the data so that there are records with a species label that were not seen during pretraining. However, our simulation does not necessarily match the real-world distribution of truly undiscovered species.

To truly investigate the usefulness of CLIBD, it is necessary to work with experts in biomonitoring in live-deployment, which is currently outside the scope of this work.

## D.2   WORKFLOW FOR SPECIES IDENTIFICATION

The use of DNA barcodes for fine-grained identification is highly accurate. However, no automated method is perfect and thus it is coupled with human verification of morphological features and inspection of the specimen. In a typical workflow, a machine learning algorithm is used to assign family and order to a new specimen, and barcodes are used to propose fine-grained BINs for the specimen. If there are inconsistencies between initial order identification and the proposed BIN, then human experts inspect the specimen and attempt to provide the correct taxonomic label. However,

not all specimens have a well-established species that can be clearly associated with it, in which case the BIN serves as a placeholder until biologists come to a consensus on whether a new species is warranted or whether the specimen (or group of specimen with the same BIN) does indeed belong to a existing species. In BOLD [51], one of the largest repositories of DNA barcodes, and the source of the BIOSCAN-1M and BIOSCAN-5M datasets, there are 1.2M BINs but only 259k named species.

We hope that CLIBD can provide a multimodal model that can eventually be integrated into such a workflow, to provide potentially accurate classification, or tools for biologists to retrieve similar samples via either image or DNA. However, the study of how to integrate our model into the workflow is beyond the scope of this this work.

## D.3 ACQUISITION COST

Compared to acquiring taxonomic labels, DNA barcoding is an automated process that provides an identifying DNA sequence (e.g. the DNA barcode) per specimen. The Centre for Biodiversity Genomics (CBG), which contributed the BIOSCAN data, has been developing protocols that allow for barcode analysis for under $1. They are continuing to develop cheaper, less invasive DNA barcoding technology, with the goal of driving costs down to below $0.01 per specimen. Metabarcoding (the barcoding of multiple specimens at once) is even more promising from a cost perspective. CBG is now targeting Oxford Nanopore Technology flow cell-based protocols that will lower the sequencing cost for barcode acquisition from $0.05 to $0.001.

On the other hand, obtaining taxonomic labels is currently a human driven process which can cost an average of at least $25 per sample. Depending on whether the species is known, the detail in the image, and potential dissection involved, a taxonomic specialist may require a few minutes to often a couple hours to identify a sample. There are many specimens that are visually similar, and most specimens are not labelled to fine-grained taxa like genus or species because of ambiguities, expert disagreement, or lack of established labels.

## D.4 COMPUTATIONAL AND STORAGE COSTS

In addition to annotation costs, it is important to consider computational and storage costs. These costs are directly related to the length of the DNA sequence. In this work, we use DNA barcodes from the mitochondrial COI gene, which have a length of 660 base pairs (bp).

**Token length.** For the tokenization methods we use (5-mer non-overlapping tokens for DNA barcodes and WordPiece for taxonomic labels), we obtain approximately 133 tokens for DNA and 20 for taxonomic labels.

**Computational cost.** Validating on a batch of 500 samples, we find that the number of FLOPS for BarcodeBERT (87.4M parameters) is 5.90T, compared to 0.13T FLOPS for LoRA-BERT (29.2M parameters) for taxonomic labels. Despite the difference in FLOPS, we find that the runtime per batch is nearly equivalent for both models at 15 ms per batch.

**Storage cost.** Without compression, we find that the BIOSCAN-1M samples with species labels (84,450) require 53 MB of storage for DNA barcodes vs. 12 MB for taxonomic labels. Given that DNA consists primarily of four nucleotides (A, T, C, and G), a simple compression of 2 bits per nucleotide yields a size of 13.3 MB for the DNA barcodes.

