# OpenReview forum: "CLIBD: Bridging Vision and Genomics for Biodiversity Monitoring at Scale"
_ICLR.cc/2025/Conference — ICLR 2025 Poster_

### Official Review · Reviewer_JVaW · 2024-10-25

**Soundness:** 1
**Presentation:** 3
**Contribution:** 2
**Rating:** 3
**Confidence:** 4

**Summary:**

The paper proposes CLIPBD, a tri-modal embedding space consisting of image, text and DNA of insect specimens. CLIPBD is trained using a three-way contrastive learning objective between image, text and DNA on the BIOSCAN-1M dataset. Using a reference database of images and DNA, CLIPBD is able to classify images of seen and unseen species. Results reported in the paper show superior performance as compared to BioCLIP and other state-of-the-art DNA encoder models.

**Strengths:**

- The motivation and problem formulation is sound and interesting.
- The proposed model fuses images, text and DNA into a contrastive embedding space enabling zero-shot image classification of unseen species.
- The flow of paper and writing is good in general.

**Weaknesses:**

- The abstract claims the paper is the first to use contrastive learning to fuse DNA and image. However, there are existing works [1, 2, 3] which have done this for other applications and they should be discussed in the related works.
- The claim that *DNA is a better target than taxonomic labels* (**Line 316**) is highly questionable. This claim is reiterated in **Lines 359-362**. The paper clearly mentions that majority of the BIOSCAN-1M dataset does not have taxonomic labels. In fact only 3.36% pretraining data has labels upto the species level (**Line 321**). It is clearly seen from **Table-1** that aligning with taxonomic labels outperforms aligning with DNA at the order level. I believe if the authors used an **unbiased dataset** containing the same proportions of DNA labels and taxonomic labels, the results would have been similar if not worse. This is more of a problem of the dataset and not the modality itself.
- Following the previous point, why does Image-to-DNA retrieval performance improve at the species level when aligning all three modalities (**Table 1**)?
- For inference to work on unseen species during training, the framework assumes that their DNA and/or images are available in the lookup database. This is an unrealistic assumption. If the images and DNA are already available for unseen species, they might as well should have been used for training.
- Limited technical novelty considering no new representation learning technique has been proposed. The paper uses an existing dataset containing 1M insect specimens and unbalanced DNA and taxonomic labels, raising questions on the effectiveness of the method on other real-world datasets. Majority of the experiments and evaluations are only shown for a single dataset.
- The authors correctly pointed that BioCLIP was trained on diverse set of species including natural images. However, one easy way to utilize the BioCLIP embedding space would be to align the DNA modality with frozen BioCLIP vision and text encoders. Have the authors compared their method with this ImageBind-style training?


References

[1] Taleb, Aiham, et al. "Contig: Self-supervised multimodal contrastive learning for medical imaging with genetics." Proceedings of the IEEE/CVF Conference on Computer Vision and Pattern Recognition. 2022.

[2] Xie, Ronald, et al. "Spatially Resolved Gene Expression Prediction from Histology Images via Bi-modal Contrastive Learning." Advances in Neural Information Processing Systems 36 (2023).

[3] Min, Wenwen, et al. "Multimodal contrastive learning for spatial gene expression prediction using histology images." arXiv preprint arXiv:2407.08216 (2024).

**Questions:**

Details about the reference database is missing. How many images and DNA barcodes are present in the reference database during inference?

---

> ### Author Response · Authors · 2024-11-20
>
> Thank you for your review. In addition to the general response, we address your specific comments below:
>
> **First use of contrastive learning to fuse DNA and image**: We thank you for pointing this out and highlighting relevant related works from the medical imaging literature which utilize human DNA. We were not aware of these works. We adjusted the language in the abstract to highlight the novelty of fusing barcode DNA and images, and we have added a subsection at the end of the related works section to discuss the works you pointed us to. While similar in their applications to DNA, we believe barcodes have a unique function as mitochondrial DNA for ecology and taxonomy compared to gene expression and other functionality for nuclear DNA, thus situating the trajectory of our work to extract information unique to barcodes.
>
> **DNA versus taxonomic labels as a target**: We apologize for the confusion. Firstly, to clarify, BIOSCAN-1M has taxonomic labels for all samples, but only 3.36% of the pretraining data is labeled up to the species. For all experiments, we use all known taxonomic labels during training.
> That notwithstanding, we agree with the reviewer that, if all samples had species labels, and given the objective of maximizing performance at species classification, one would most likely fare better by training that model on species labels than on barcode DNA, since it is precisely our evaluation task. However, this is not the situation we find ourselves in. DNA barcodes can be acquired for a cost of less than \\$1 per specimen (with costs per specimen decreasing year on year) and can be deployed at scale, whereas manual labelling costs upwards of \\$25 per specimen, and expert annotators have limited availability that can not be scaled even if funding permitted it. As a consequence of this, we are able to train our model on BIOSCAN-1M, with 1M paired samples of images and DNA barcodes, whereas a dataset of 1M insect samples labelled to species level simply does not exist. Fundamentally, this is not "a problem of the dataset" but more an issue of the practicalities of data collection and annotation at scale. Hence, conditioned on the data that is available now and for the foreseeable future, it is better to use the modality which scales better (whilst also containing information to species-level granularity), and our work and results demonstrate the success of this approach. As the reviewer points out, we are upfront that the reason DNA barcodes outperform species labels is because species labels have sparse availability. We have adjusted the text further to clarify our claims on this point, in line with this discussion with the reviewer.
>
> **Improvement at species level when aligning all three modalities**: We believe that this observation is precisely one of our major claims, that cross-modal alignment can help improve alignment in the other modalities. Thus, we do believe that alignment against taxonomic labels can in certain cases benefit even image-DNA alignment. However, while all three modalities provide value, we believe that DNA is more beneficial than text as a supervisory modality for images because of its scalable collection, robustness to error and expert disagreement, and contribution to performance.
>
> **Realism of having DNA/images of unseen species as keys**: Unseen species refers to species not seen during the training of the model, rather than species being discovered for the first time when passed to the model. While images and DNA may be available, retraining the model every time a new species is discovered and labelled is unlikely to be a feasible solution. It is estimated that around 80% of insect species are as yet undiscovered [1]; correspondingly, insect biodiversity surveys are constantly encountering new species, and thus it is very realistic to encounter this scenario frequently. Whilst it may be practical to retrain the model on more data periodically (after collecting multiple samples of multiple new species), we explore a solution which can operate in the intervening period, where a new species has been observed and labelled but the model has not yet been trained on it. Additionally, specimens are commonly barcoded/imaged and assigned an operational taxonomic unit (e.g. by binning) but not immediately given a scientific name. It may be preferable for the model not to be trained on these specimens until formally recognized but to still be able to match embeddings to them. In these cases, we envisage adding the embeddings of the new samples (both of new species and known species) to the vector database to use as keys for retrieval. This is the scenario we simulate by splitting out seen and unseen species, and dividing the samples into keys and queries.
>
> [1] Stork. Ann Rev Entomology, 2018. doi:10.1146/annurev-ento-020117-043348
>
> **Methodological novelty**: Please see our general response for the novelty of our work.

---

> ### Author Response · Authors · 2024-11-20
>
> (cont. from previous post)
>
> **Majority of exps and evals are only shown for a single dataset**: We were limited to only using BIOSCAN-1M because there aren’t other comparable insect datasets with both images and barcodes available. Where possible, we evaluated on additional datasets such as the INSECT dataset (Table 4) to demonstrate the generalizability of our method.
>
> **Aligning DNA embeddings with BioCLIP space**: We thank the reviewer for their suggestion. We will update the paper with the experiment results of aligning the DNA modality with frozen BioCLIP encoders as this will enable a useful comparison on Image-to-DNA lookup against our CLIBD models. Of course, with such an experimental setup the Image-to-Image and Image-to-Text performance will remain unchanged from the results we report in Table 2, since the image and text encoders are frozen.
>
> **Reference database size**: The reference database used during evaluation is the key split referred to in Figure 2, comprising 21,118 samples, each of which has an image and a barcode. We have updated the caption of Figure 2 to reflect this. We also added Table 5 to the appendix to share further statistics of the dataset, including the number of specimens with each label and the number of labels in each split.

---

> ### Comment · Reviewer_JVaW · 2024-11-21
>
> I thank the authors for improving the writing and conducting additional experiments. I will still stay with my original rating of **3: Reject**. The reasons are as follows:
>
> 1. Making claims about DNA is a better modality than the text modality just from the perspective of *cost of annotation* and not from an *information-theoretic* or *computational* sense is obscure, given the poor quality of taxonomic labels in BIOSCAN-1M.
> 2. DNA sequences can be very long, while taxonomic labels are only a few tokens. What about the computational difference (in FLOPS) between processing the DNA vs the taxonomic labels using a BERT-like model? Have you considered that cost?
> 3. What about the storage *cost* difference (in GB) between storing a database of long DNA sequences and taxonomic labels?
> 4. Taxonomic hierarchies are not inherently encoded in DNA and are not easier to extract. While taxonomic labels provide a more intuitive medium to understand a species type for a *non-expert user*.
> > It is estimated that around 80% of insect species are as yet undiscovered.
>
> 5. In the real world, how do you decide which species is undiscovered from the model? Some human-expert annotation is still required to verify a species not present in the training set of the model.
> 6. Technical novelty is still not clear.
> 7. The overall story and conclusions are unclear considering the points mentioned earlier.

---

> > ### Author Response · Authors · 2024-11-22
> >
> > Thank you for the quick reply. We apologize that we did not make clear in our original response that we are working with **DNA barcodes**, short snippets of DNA that are informative of the species.  In particular, “DNA barcoding can increase taxonomic resolution and harmonize the identification of taxa which are difficult to identify or lack experts” (see https://en.wikipedia.org/wiki/DNA_barcoding#Potentials)
> >
> > While in general DNA sequences can be very long (the average human gene is 10-15 kilo base pairs [2]), **DNA barcodes** are short, standardized gene sequences ranging in length from 400 to 800 base pairs (bp) and selected specifically for evolutionary stability and species identification [1]. For insects and other animals, scientists use the mitochondrial COI gene, with a length of 660 bp (see L33-35 and L254-255). Given that the DNA barcode is not long, the storage and compute requirements to use it are not especially demanding. Below we provide responses to the reviewer’s specific questions.
> >
> > > 1. Making claims about DNA is a better modality than the text modality just from the perspective of cost of annotation and not from an information-theoretic or computational sense is obscure, given the poor quality of taxonomic labels in BIOSCAN-1M.
> >
> > We agree that it is important to consider other aspects such as the information content and computational cost of using DNA barcodes.  In this work, we want to demonstrate that DNA barcodes are a feasible and useful source of information for taxonomic classification.  We believe that it is potentially better than taxonomic labels in terms of annotation cost and information content and that it does not have high computational overhead.
> > In addition, our argument about the usefulness of DNA barcodes is not meant to assert that future work should only use one or the other but rather to compare their effectiveness. We actually advocate for the use of taxonomic labels in training when available.
> > We base our claim on the following:
> >
> > 1. **Cost of annotation**: DNA is cheaper and faster to acquire than taxonomic labels.
> > 2. **Objectiveness**: Taxonomy labels are subjective due to expert disagreement or ambiguity. In many cases, experts are unable to annotate down to the species level. DNA barcodes, on the other hand, are objective and more robust to error.
> > 3. **Resolution**: In cases where organisms are morphologically similar, DNA barcodes are used to help experts resolve their taxa, especially below the family taxon, and make fine-grained annotations.
> > 4. **Better performance**: A valid comparison of performance can either occur under an equal number of labels or an equitable number (fair for the relative numbers of labels we can reasonably expect to have). Since it is infeasible to evaluate under the former, given the low number of species labels, we evaluate under the latter scenario, for which we find that aligning against DNA leads to better image-image accuracy than aligning to text.
> > 5. **Predictive power**: While taxonomy labels are primarily limited to species identification, DNA barcodes encode significantly richer information about the evolutionary development of species. Research has demonstrated the ability to use DNA barcodes to model how species evolve into communities and have used them to comprehend the evolutionary and taxonomic tree [1].
> >
> > For taxonomic classification specifically, species labels comprise a more direct alignment target. However, DNA barcodes are still highly correlated with species and sufficient for extracting taxonomic information, as shown by our experimental results. While they are slightly more computationally expensive than taxonomy labels, the additional cost is relatively insignificant due to their short length, as we show below.
> >
> > > 2. DNA sequences can be very long, while taxonomic labels are only a few tokens. What about the computational difference (in FLOPS) between processing the DNA vs the taxonomic labels using a BERT-like model? Have you considered that cost?
> >
> > For the tokenization methods we use (5-mer non-overlapping tokens for DNA barcodes and WordPiece for taxonomic labels), we obtain approximately 133 tokens for DNA and 20 for taxonomic labels. Validating on a batch of 500 samples, we find that the number of FLOPS for BarcodeBERT (87.4M parameters) is 5.90T, compared to 0.13T FLOPS for LoRA-BERT (29.2M parameters) for taxonomic labels. Despite the difference in FLOPS, we find that the runtime per batch is nearly equivalent for both models at 15 ms per batch. While we did consider computational cost, we did not emphasize it because the computational burden of processing DNA barcodes is relatively insignificant for the benefits.

---

> > ### Author Response · Authors · 2024-11-22
> >
> > (cont. from prev post)
> >
> > > 3. What about the storage cost difference (in GB) between storing a database of long DNA sequences and taxonomic labels?
> >
> > Without compression, we find that the BIOSCAN-1M samples with species labels (84,450) require 53 MB of storage for DNA barcodes vs. 12 MB for taxonomic labels. Given that DNA consists primarily of four nucleotides (A, T, C, and G), a simple compression of 2 bits per nucleotide yields a size of 13.3 MB for the barcode DNA.
> >
> > Furthermore, the embedding sizes for the reference database would be the same for either modality.
> >
> > Note also that for biodiversity efforts in which DNA barcodes are collected, they are an essential part of the specimen data and will be stored alongside taxonomic labels, regardless of which modality is used during alignment.
> >
> > > 4. Taxonomic hierarchies are not inherently encoded in DNA and are not easier to extract. While taxonomic labels provide a more intuitive medium to understand a species type for a non-expert user.
> >
> > We agree that taxonomic labels are more human interpretable, and it is precisely why we aim to predict these taxonomic labels. However, the predictive target does not need to be identical to the alignment target used during training. We argue that DNA barcodes are a better modality as an alignment target as they are *richer and more objective* than taxonomic labels, encoding evolutionary information and species development unlike taxonomic labels. We show empirically that we can learn good representations without comprehensive taxonomic labels.
> >
> > > 5. In the real world, how do you decide which species is undiscovered from the model? Some human-expert annotation is still required to verify a species not present in the training set of the model.
> >
> > We treat all species at inference time the same and do not need to determine whether a species is “undiscovered”. The distinction is made primarily for the purposes of evaluation. No human-expert annotations are required during inference time.
> > In Section 5.4, we do investigate whether we can determine whether a new sample corresponds to the set of “seen” species that the model was trained on or not, simulating unseen species as a proxy for novel species detection. We plan to explore this further in future work.
> >
> > > 6. Technical novelty is still not clear.
> > > 7. The overall story and conclusions are unclear considering the points mentioned earlier.
> >
> > The main goal of our paper is to investigate whether *aligning to DNA barcodes yields good representations for taxonomic classification*. Prior work for taxonomic classification and evolutionary study has only explored aligning images with taxonomic labels, whereas DNA barcodes are easier to scale and more predictive for downstream tasks, including taxonomic classification but also potentially other problems in ecology. Leveraging DNA barcodes in contrastive learning to build effective embedding representations has not been achieved prior, and we believe that we have successfully demonstrated its viability for future work.
> >
> > We hope that we have helped clarify the confusion regarding the specific type of DNA that we use and have provided information about the value and rationale of using DNA barcodes. Please let us know if you have additional questions.
> >
> > [1] Kress, Erickson. Methods Mol Biol. 2012;858:3-8. doi:10.1007/978-1-61779-591-6_1.
> >
> > [2] Tom Strachan and Andrew P. Read. Human Molecular Genetics, 1999 Garland Science.

---

> > > ### Comment · Reviewer_JVaW · 2024-11-22
> > >
> > > I thank the authors for providing a comprehensive set of clarifications. While I agree to some, I disagree with most. The disagreements are as follows:
> > >
> > > > Taxonomy labels are subjective due to expert disagreement or ambiguity. In many cases, experts are unable to annotate down to the species level. DNA barcodes, on the other hand, are objective and more robust to error.
> > >
> > > 1. I disagree to some extent. Several studies have highlighted errors in the DNA barcoding process and misidentification of species due to DNA barcoding. For example [1] highlighted a misidentification rate of ~17% in cowries. [2] conducted a systematic evaluation of DNA barcoding process of insects and concluded that errors in barcode data is not rare and most commonly due to human. In this work, you use a "high-quality toy dataset" to demonstrate the performance of the models, which does not exist in a live production environment.
> > >
> > > > No human-expert annotations are required during inference time.
> > >
> > > 2. I completely disagree with this. After your model is deployed in the real world and it starts making predictions, how do you detect **failure** modes? How do you detect the model is not making highly confident false predictions? Expert biologists are still required to verify the outputs from the model. This situation will get worse if the model is provided with an image of species that was not in the training set.
> > >
> > > References
> > >
> > > [1] Meyer, Christopher P., and Gustav Paulay. "DNA barcoding: error rates based on comprehensive sampling." PLoS biology 3.12 (2005): e422.
> > >
> > > [2] Cheng, Zhentao, et al. "The devil is in the details: Problems in DNA barcoding practices indicated by systematic evaluation of insect barcodes." Frontiers in Ecology and Evolution 11 (2023): 1149839.

---

> > > > ### Author Response · Authors · 2024-11-23
> > > >
> > > > Thank you for your questions!  We appreciate you taking the time to read over our responses and following up with additional questions and discussion.
> > > >
> > > > > 1. I disagree to some extent. Several studies have highlighted errors in the DNA barcoding process and misidentification of species due to DNA barcoding. For example [1] highlighted a misidentification rate of ~17% in cowries. [2] conducted a systematic evaluation of DNA barcoding process of insects and concluded that errors in barcode data is not rare and most commonly due to human. In this work, you use a "high-quality toy dataset" to demonstrate the performance of the models, which does not exist in a live production environment.
> > > >
> > > > In our claim, we were attempting to state that DNA barcode is a property of the organism.  It is objective in that there is one definitively true sequence. In contrast, taxonomic labels are designed by humans.
> > > >
> > > > Fundamentally, we agree with the reviewer’s point that when used for taxonomic classification, relying solely on DNA barcodes is imperfect.
> > > >
> > > > We agree that evaluating on a fixed dataset is not the same as deploying in a live environment. Our goal is to stimulate interest and research in this area. By benchmarking on a fixed dataset, we can demonstrate the efficacy of our approach. We hope that there will be follow-up work that investigates how our model can be used in a live deployment (we hope to collaborate with biodiversity experts to explore this), as well as research that studies different (potentially better) models for taxonomic classification.
> > > >
> > > > > 2. I completely disagree with this. After your model is deployed in the real world and it starts making predictions, how do you detect failure modes? How do you detect the model is not making highly confident false predictions? Expert biologists are still required to verify the outputs from the model. This situation will get worse if the model is provided with an image of species that was not in the training set.
> > > >
> > > > Our claim about annotations during inference was about inference in our experiments, for which the model only produces its prediction based on the raw image or DNA input query.
> > > >
> > > > For a production deployment of the model, we agree that a human-in-the-loop system would be necessary to validate the ongoing performance of the model and identify areas for improvements. A combination of model outputs, such as using the method investigated in Section 5.4, combined with human annotation and validation, could be leveraged to identify new species. Over time, as the performance of the model is proven and methods of identifying when predictions may be wrong are developed, our goal would be to reduce the percentage of cases which humans need to validate and thus the overall annotation burden.
> > > >
> > > > Taking a step back, we would like to reiterate that our contribution is not the study of a live deployment or “production” deployment of a model. Our contribution is the study of whether using contrastive training that aligns representations of DNA barcodes to images, we can obtain representations that are useful for taxonomic classification. We believe our experiments on the BIOSCAN-1M datasets demonstrate that DNA barcodes are a useful signal for alignment.
> > > >
> > > > We also agree that this study is limited in that: 1) we are evaluating on a static dataset and using it to simulate different scenarios, and 2) that a live deployment of the model will likely involve additional innovations and human-in-the-loop feedback.
> > > >
> > > > We are happy to incorporate a more detailed discussion of the above limitations in our paper, if the reviewer would like.

---

> ### Comment · Reviewer_JVaW · 2024-11-23
>
> I thank the authors for the clarifications and taking time to address each of my concerns. As I mentioned in my *original review*, I liked the problem motivation and completely agree that DNA barcodes can prove to be an important modality. However, the current state of the paper is such that the experimental setup and writing needs to be refined a lot to be able to make any strong conclusions. This makes the overall story blurry as the readers will be left with a lot of unanswered questions after reading the paper. As a result, i am reluctant to increase my score.

---

> > ### Author Response · Authors · 2024-11-27
> >
> > We would like to thank you for carefully reading our paper and engaging in discussion.  Following your suggestions, we have tried to revise our paper so that we more clearly state the role of barcode DNA in our work.
> >
> > > However, the current state of the paper is such that the experimental setup and writing needs to be refined a lot to be able to make any strong conclusions
> > Thank you for your feedback.  Overall, reviewers had indicated that our paper “well-written and and organized” (Ujh5,CfWv), and that the “proposed approach and the experiments are clearly described” (Ujh5)
> >
> > Nevertheless, we have attempted to further polish and refine our text so that the pertinent information is stressed clearly.  We have especially paid attention to questions raised during our discussion so that certain points are included in the revised draft.
> >
> > Below we summarize some updates we made in response to your original set of questions:
> > > abstract claims the paper is the first to use contrastive learning to fuse DNA and image
> > - We have clarified the abstract to indicate that we focus on barcode DNA and added discussion of suggested works to the end of the related works section.
> >
> > > The claim that DNA is a better target than taxonomic labels (Line 316) is highly questionable.
> > - We have toned down the claim to make it specific to BIOSCAN-1M (which was the intent as the experiment was conducted on BIOSCAN-1M).
> > - We have also added a discussion of the limitations and advantages of barcode DNA to Appendix D. We ask you to have a look at this section as it was influenced heavily by our discussion with you through the review process.
> >
> > > Have the authors compared their method with this ImageBind-style training?
> > - We have added ImageBind-style training to Appendix B.1.1
> > - We compared ImageBind and CLIBD as two different alignment methods. In ImageBind, the image and text encoders are frozen, aligning only the DNA encoder with the image encoder. In CLIBD, we align all three encoders to each other. We also experimented with binding BarcodeBERT to both the BioCLIP image and text encoders. The results show that simultaneous alignment with both image and text encoders yields marginally higher performance compared to binding solely with the image encoder. However, both approaches underperform our proposed CLIBD. This underscores the benefits of our approach's comprehensive alignment strategy. We note that since these experiments utilized BioCLIP's pretrained models, which were trained on BIOSCAN-1M data, the evaluation of unseen species may be influenced by potential exposure during pretraining.
> > This analysis strengthens the case for CLIBD's design choices and demonstrates that maximizing the learning potential across all modalities through simultaneous alignment produces superior results compared to more constrained training paradigms.
> >
> > Please let us know if you have any specific concerns or parts of the paper that you find unclear.
> >
> > We sincerely appreciate you engaging with us and appreciate the time you have put not just into your review but in the ongoing discussion. There are two remaining points that we’d like to address:
> >
> > **Choice of dataset**: We agree with the reviewer that our work is limited to just BIOSCAN-1M and the INSECT datasets.  We chose them as they were datasets that included both DNA barcodes and images.
> >
> > We respectfully disagree with the reviewer’s characterization of BIOSCAN-1M as a "high-quality toy dataset".  This dataset was contributed by the [Centre for Biodiversity Genomics (CBG)](https://biodiversitygenomics.net/) and reflects the real-world challenges of identifying and providing a taxonomic label to every specimen. Through communications with the CBG, we learned that the records that lack species labels result from a conscious decision by the curators to not provide labels for records for which they were not confident, to minimize the impact of human error. We believe working on BIOSCAN-1M will allow us to build models that may in the future be incorporated into a production workflow for species identification.  However, that is out of scope for our current work.
> >
> > **Novelty**: We agree with the reviewer that our work is not focused on contributing new technically novel methods.  However, we want to point out that contributions to the community come in different forms, including providing empirical experiments and evidence.  Our work shows that by using information from DNA barcodes with images to train our models, we can obtain improved representations.  By using taxonomic labels, the classification accuracy of our model further improved.  We believe that this combination of modalities is a fruitful direction for further research, and can stimulate in some members of the ICLR community an interest in DNA barcodes and encourage more work on applications of ML techniques for biodiversity monitoring.

---

### Official Review · Reviewer_CfWv · 2024-10-26

**Soundness:** 3
**Presentation:** 4
**Contribution:** 3
**Rating:** 6
**Confidence:** 4

**Summary:**

This paper proposes a CLIP-based method to jointly embed images, DNA barcodes, and taxonomic strings for different insect species. The method is evaluated on cross-modal matching and species classification, with qualitative and quantitative results provided. The paper compares against one external method (BIOCLIP) in one set of experiments and one external method (BZSL) in a second set of experiments. The key claim of the paper is that jointly learning from the modalities of images, DNA, and taxonomic information leads to stronger representations for downstream use.

Note: Score increased after rebuttal.

**Strengths:**

* The idea of jointly embedding DNA barcodes with images and taxonomic information is interesting.
* The experiments in the paper are extensive - there is a lot of technical content, and it's clear that a lot of effort went in to this work. There are many quantitative results, in addition to interesting qualitative results (e.g. Fig. 3, Fig. 5).
* The paper is very well-written.
* The hyperparameters and training procedures are clearly spelled out.

**Weaknesses:**

My major issue with the paper is missing baselines:
* The paper compares their multimodal representation learning approach against the unimodal pretrained models they start with. They show that their method is better, and conclude that multimodality is important. However, this claim is not justified - couldn't the benefit be from the additional training each modality received? It seems to me that the fair comparison would be to take the unimodal models and run unimodal CLIP-style training (with a similar computation / # steps budget) for each. If the multimodal model beats these CLIP-fine-tuned unimodal models, then that provides stronger evidence of the benefit of multimodal learning.
* For the image encoder, wouldn't a model pretrained on BIOSCAN-1M be a more appropriate starting point / baseline than an ImageNet-pretrained model?

The paper should address a few items that were not discussed:
* Are there confounders we need to worry about in this data, e.g. the facility the data was collected by?
* The DNA to DNA matching results are very high, but shouldn't we expect this? Would simple homology methods based on string matching do a very good job at this task? What is the benefit of using deep embeddings to solve this problem?
* Doesn't adding a modality increase the number of steps of training the model receives? Couldn't this be partly responsible for the differences between 1, 2, and 3 modalities in Table 1?
* Image-DNA and Image-text matching don't seem very good based on Fig. 4, with the average being pulled up by a few good cases. What are those cases, and why are they different? Is there any insight to be gained there?

A few claims made in the paper were not clear to me:
* The paper claims that "BIOCLIP... requires taxonomic labels to be available in order to obtain text descriptions. These labels can be expensive and time-consuming to obtain" and that "DNA barcodes can be obtained at scale more readily than taxonomic labels". These claims are not intuitive to me. How much does DNA barcoding cost per individual, compared to the cost of having an expert inspect an image to identify the species?
* The training data for BarcodeBERT is claimed to be "different from, but highly similar to" the data used in this paper  - can you expand on what this means, and the implications for the results presented?

**Questions:**

Please see weaknesses for primary questions.

Minor comments / questions (no need to respond):
* This work focuses on the cases where aligned data is available: images and DNA barcodes from the same individuals. It might be useful to extend the method to also take advantage of abundant unpaired data: images and DNA barcodes that are not paired.
* Figure 2 could be clearer. Why are there different shades of blue and orange? Why don't the numbers in the boxes sum up to 36729? Generally, this figure did not aid my understanding (though many of my confusions were clarified in later text and figures).
* It would be nice to have a chance-level baseline in Table 1.
* Down the road, it might be interesting to try to integrate additional modalities, e.g. geospatial location (https://arxiv.org/abs/1906.05272).

---

> ### Author Response · Authors · 2024-11-20
>
> We sincerely thank the reviewer for their probing questions, as these highlighted several points upon which it would be useful to adjust the manuscript, improving its clarity and comprehensiveness.
>
> **Unimodal CLIP-style baseline**: We note that the CLIP training objective is an inherently multimodal contrastive learning objective, so the methodology can not be directly transferred to unimodal experiments. Nevertheless, we agree with the reviewer that we need to have unimodal pretraining on the dataset to demonstrate that the improvement in performance comes from leveraging the multimodal training, not just adapting to the dataset domain. To address this question, we ran unimodal SSL experiments following the SimCLR methodology and found that it was still outperformed on image-image matching by the models trained with multimodal contrastive learning, achieving 10.9% macro accuracy for species compared to 51.2% by the model trained with image and DNA. We have added the full results to Tables 1 and 8 through 10 in the paper. We will also add results of training CLIBD with an image encoder pretrained on BIOSCAN-1M.
>
> **Pretrained image encoder on BIOSCAN-1M**: We believe that the general visual knowledge from an ImageNet-pretrained model can still be useful but agree that starting from a model pretrained on BIOSCAN-1M could be beneficial. We will add experiment results for CLIBD using the SimCLR-trained image encoder for initialization to the upcoming revision.
>
> **Dataset confounders**: While the specimens in BIOSCAN-1M are collected by different teams in three countries—Canada, South Africa, and Costa Rica—they are imaged and sequenced at one central facility, the Centre for Biodiversity Genomics (CBG). CBG employs rigorous protocols to ensure data quality. If the reviewer has more specific concerns, we would be happy to add details. We did not provide a lot of text concerning the data pipeline as it has been published in the BIOSCAN-1M paper (NeurIPS 2023).
>
> **DNA-to-DNA matching**: We found that simple homology methods such as BLAST achieve higher DNA-to-DNA performance compared to contrastive learning when run on the test queries against the full key set at 99.2% macro accuracy (computed with the harmonic mean between seen and unseen), but the runtime is significantly longer (9.3 minutes to run BLAST on the unseen test samples compared to 1.1 minutes to run CLIBD and query embeddings). In addition, having dense DNA barcode embeddings can be easily incorporated into neural networks for other downstream tasks, including evolution and specialization of species in addition to species identification [1].
>
> [1] Kress, Erickson. Methods Mol Biol. 2012;858:3-8. doi:10.1007/978-1-61779-591-6_1.
>
> **Impact of adding modalities on number of iterations**: For all models, the number of samples used is the same and thus (with the same batch size used throughout the experiments), the number of training steps is the same. For the taxonomic labels, we concatenate the labels (order, family, genus, and species) up to the finest level that is known. So if the species label does not exist, we just include the labels up to the genus level. With more modalities, more aspects of the data are considered in the loss so the loss term has a larger magnitude, but this is only useful in as much as these additional contributors to the loss contain informative training signals. This is what we are seeking to investigate as the premise of these experiments.
>
> **Cross-modal matching performance trends**: We account for outlier species by measuring performance by macro accuracy, with equal weight among species, rather than only micro accuracy, in which species with many samples would dominate performance. We have clarified Figure 4 to represent the proper density of species across the range of sizes and show that there are only a few outlier cases with high performances, thus having a relatively low impact on the macro accuracy.
> For seen species, performance increases for species with more key samples. For unseen species, image-text matching is low because the text model has not seen any of the species names, hence we do not expect good alignment for these species. For those outliers such as Xylosandrus Morigerus, in nearly all cases, the species is the most common within its genus in the dataset, thus likely resulting in better performance by extending the alignment at higher-level taxa to the lower ones. In other cases of species such as Psychoda sp. 11GMK with many samples for which the model achieves poor performance, they do not comprise a majority of the samples in the genus within the dataset or they are the only species in the genus, in which case the model has not seen the genus before either. Ultimately, this highlights a need for further improvements to cross-modal alignment and investigation into label imbalance at training and inference time, as well as understanding and exploiting further the hierarchical nature of taxonomy.

---

> ### Author Response · Authors · 2024-11-20
>
> (cont. from previous post)
>
> **Cost of taxonomic label vs barcode acquisition**: DNA barcoding is an automated process using specialized equipment, with the current cost below \\$1 per specimen. Ongoing work is continuing to develop cheaper, less invasive DNA barcoding technology, with the goal of driving costs down to below \\$0.01 per specimen.
> For example, the BIOSCAN project proposed to develop protocols allowing barcode analysis for \\$1 and to use this capability to analyze 10M specimens to advance species discovery.
> CBG achieved the \\$1 target in late 2022, allowing more than 3M specimens to be barcoded in 2023 and 2024. Metabarcoding (the barcoding of multiple specimens at once) is even more promising from a cost perspective. CBG is now targeting Oxford Nanopore Technology flow cell-based protocols that will lower the sequencing cost for barcode acquisition from \\$0.05 to \\$0.001. On the other hand, obtaining taxonomic labels is currently a human driven process which can cost an average of at least \\$25 per sample. Depending on whether the species is known, the detail in the image, and potential dissection involved, a taxonomic specialist may require a few minutes to often a couple hours to identify a sample. There are many specimens that are visually similar, and most specimens are not labeled to fine-grained taxa like genus or species because of ambiguities, expert disagreement, or lack of established labels.
>
> **BarcodeBERT pretraining data**: BarcodeBERT is pretrained on a dataset of 1.5M barcode DNA samples from Canadian vertebrates also using the COI gene. There are no common species or barcodes between this dataset and BIOSCAN-1M, so there is no concern of data used for both BarcodeBERT pretraining and CLIBD evaluation. This pretraining dataset of barcode DNA is more similar to our data than that used for other encoders invariably pretrained on human DNA (e.g. DNABERT[-2/S], HyenaDNA), making BarcodeBERT an appropriate encoder for initialization. We have clarified the distinction in the paper and have cited the [Canada 1.5M dataset](https://www.nature.com/articles/s41597-019-0320-2).
>
> **Training with unpaired image and DNA**: Thank you for your suggestion! We agree that this would be an excellent direction for future research.
>
> **Clarity of Figure 2**: Please see the general response. In particular, the lines linking the numbers in the boxes indicate that the samples of the seen splits for train, val, and test have the same species, not that they are the same samples. We have clarified this figure, including the colors and the counts, in the latest revision.
>
> **Chance-level baseline in Table 1**: For Image-to-DNA matching, the non-aligned results are effectively chance level. To augment this result, we added Table 7 to Section B of the appendix to show the performance of the random baseline explicitly. In addition, for the micro-averaged baseline, we also included a comparison against a simple majority-label prediction, where we select the most frequent order, family, genus, species label.
>
> **Adding geospatial modality**: Thank you for your suggestion! We plan to investigate this in future work.

---

> > ### Comment · Reviewer_CfWv · 2024-11-25
> > **Response to authors**
> >
> > I thank the authors for their response and updates to the paper.
> >
> > Their response addressed my primary concerns (missing baselines) and clarified a number of my secondary concerns. While there are always areas for improvement, these changes are sufficient for me to raise my score. I think the work is interesting enough and the experiments are thorough enough to be valuable to the ICLR community.

---

> > > ### Author Response · Authors · 2024-11-27
> > >
> > > Thank you for reading our revised draft and increasing your score.   We really appreciate your suggestion of adding the unimodal training on BIOSCAN-1M.  By initializing with an image-encoder pretrained using SimCLR, we were able to outperform our initial results with with the ViT-B pretrained only on ImageNet (Appendix B.1.1).
> > >
> > > We are glad that you find our work to be interesting with thorough experiments.  We believe that biodiversity monitoring is an important problem and that through our work we can inspire more ML experts to apply their knowledge and try out more advanced learning techniques to multimodal learning with DNA barcodes.   If you believe our work can be valuable to the ICLR community, we hope that you will help champion our work.

---

### Official Review · Reviewer_s416 · 2024-11-02

**Soundness:** 2
**Presentation:** 2
**Contribution:** 2
**Rating:** 6
**Confidence:** 3

**Summary:**

The approach proposes multimodal contrastive learning between image, DNA and text (taxonomic labels) as opposed to Image+TaxonomicLabel only approach followed by previous work (BioCLIP). The usage of DNA is claimed to be better because, (1) classifying unseen species would be difficult with a taxonomic-label-only model because the species name would not have been seen during training, (2) DNA is easier to obtain compared to taxonomic label which requires careful examination by human experts.

**Strengths:**

1.	Incorporation of DNA as a modality to align the image embedding against instead of text is well motivated.
2.	Extensive experiments and ablations are provided.

**Weaknesses:**

1.	The accuracy when doing image to DNA on unseen species is not quite significant although it is better than BioCLIP’s approach of doing image to text. This indicates the image encoder is still not strong enough to generate a good DNA aligned embedding just from the image. Perhaps this can improve with more data.

**Questions:**

Some suggestions and questions,

1.	Comparison with BioCLIP at different taxa levels: Going by the works claim, incorporation of DNA embeddings could help with classifying unseen species up to species level, but I’m guessing at higher taxonomic levels the BioCLIP performance should be comparable to CLIBD. It would be interesting to see a comparison.
2.	From Table 1, Image-to-DNA performance on seen species reduces going from (I+D) to (I+D+T), why is this happening? I would have expected the performance to improve.
3.	I’m also curios to know why BIOSCAN-5M was not used, considering that foundational models such as these can greatly benefit from more data.
4.	I personally felt Figure 7 was more informative to understand the data partitioning than Figure 2 (Just to consider in the future revisions).

---

> ### Author Response · Authors · 2024-11-20
>
> Thank you for your review. In addition to the general response, we address your specific comments below:
>
> **Image-to-DNA on unseen species**: We do agree that the most challenging scenario for the model is in cross-modal performance on species not seen during pretraining. Despite the fact that performance is high for same-modality queries on both seen and unseen species, cross-modal queries perform much more poorly for unseen species than seen species. This demonstrates that generalization of the cross-modal alignment is the most challenging area of this task, and we anticipate future work will build on this. We believe that more taxonomically diverse data (e.g. BIOSCAN-5M, developed concurrently with this work) could help improve this performance further and will continue to investigate better multimodal embeddings.
>
> **BioCLIP comparison at higher-level taxa**: We thank the reviewer for highlighting this. At all taxa evaluated, we observe that CLIBD still significantly outperforms BioCLIP for both image-to-image and image-to-text matching by 20 or more points. This demonstrates that even at taxa for which more samples have labels (e.g. order), we still see better performance by aligning with DNA barcodes. We have added Table 11 with the results of BioCLIP vs. CLIBD at higher-level taxa to Section B of the appendix.
>
> | Taxon. level | Model       | Img | DNA | Txt    | I-I Seen | I-I Unseen | I-I H.M. | I-D Seen | I-D Unseen | I-D H.M. | I-T Seen | I-T Unseen | I-T H.M. |
> |--------------|-------------|-----|-----|--------|----------|----------|----------|----------|----------|----------|----------|----------|----------|
> | **Order**    | BioCLIP     | ✓   | ✗   | ✓      | 73.3     | 69.2     | 71.2     | ---      | ---      | ---      | 38.6     | 35.5     | 37.0     |
> |              | CLIBD       | ✓   | ✗   | ✓      | 99.6     | 97.4     | **98.5** | ---      | ---      | ---      | **99.6** | **77.4** | **87.1** |
> |              | CLIBD       | ✓   | ✓   | ✓      | **99.7** | 94.4     | 97.0     | **99.4** | **88.5** | **93.6** | 99.2     | 74.7     | 85.2     |
> | **Family**   | BioCLIP     | ✓   | ✗   | ✓      | 56.9     | 42.2     | 48.5     | ---      | ---      | ---      | 18.9     | 14.6     | 16.4     |
> |              | CLIBD       | ✓   | ✗   | ✓      | 90.7     | 76.7     | 83.1     | ---      | ---      | ---      | 94.9     | 49.9     | 65.4     |
> |              | CLIBD       | ✓   | ✓   | ✓      | **90.9** | **81.8** | **86.1** | **90.8** | **50.1** | **64.6** | **95.8** | **50.9** | **66.5** |
> | **Genus**    | BioCLIP     | ✓   | ✗   | ✓      | 33.8     | 24.3     | 28.3     | ---      | ---      | ---      | 9.5      | 5.9      | 7.3      |
> |              | CLIBD       | ✓   | ✗   | ✓      | 72.1     | 49.6     | 58.8     | ---      | ---      | ---      | 81.9     | **81.9** | 29.7     |
> |              | CLIBD       | ✓   | ✓   | ✓      | **74.6** | **60.4** | **66.8** | **70.6**   | **20.8** | **32.1** | **83.0** | 21.6     | **34.3** |
> | **Species**  | BioCLIP     | ✓   | ✗   | ✓      | 20.4     | 14.8     | 17.1     | ---      | ---      | ---      | 4.2      | 3.1      | 3.6      |
> |              | CLIBD       | ✓   | ✗   | ✓      | 54.2     | 33.6     | 41.5     | ---      | ---      | ---      | **57.6** | 4.6      | 8.5      |
> |              | CLIBD       | ✓   | ✓   | ✓      | **59.3** | **45.0** | **51.2** | **51.6** | **8.6**  | **14.7** | 56.0     | **4.8**  | **8.9**  |
>
> (I-I, I-D, and I-T represent image-to-image, image-to-DNA, and image-to-text, respectively.)
>
> **Reduction in performance when adding taxonomic label modality**: We discussed our analysis of this in Section 5.1 ("DNA is a better alignment target than taxonomic labels") but will improve the language of the paper to make it clearer which models we were referring to. We reason that the model trained on all three modalities performed similarly or worse in certain cases at the species level due to the low proportion of samples with species labels (~3.36% of the pretraining data) and challenges with obtaining high quality species labels, as a result of expert disagreement and visual ambiguity. Note that we use all of the samples for training, but we will only have the taxonomic labels up to the most fine-grained known level to pass to the text encoder.
>
> **Use of BIOSCAN-1M over 5M**: CLIBD was developed concurrently with BIOSCAN-5M (NeurIPS 2024), which was not released at the time of our model development. We do expect the performance to improve with more data and we will further investigate the benefits of data scaling for our model.
>
> **Clarity of Figure 2**: Please see the general response. Our primary goal with Figure 2 was to represent the final split statistics rather than the partitioning process. We have clarified this figure in the latest revision.

---

> ### Author Response · Authors · 2024-11-27
>
> Thank you for your detailed review and your support of our work. Based on suggestions and feedback from you and other reviewers, we have revised our paper to include a discussion of the unimodal baseline (updated Table 1), experiments with ImageBind-style training (Appendix B.1.1), full taxonomic classification results in our BioCLIP comparison (Appendix B.1.2), additional statistics (Appendix A), a more in-depth discussion of DNA barcodes (Appendix D), and other information and clarifications (see general response for details).
>
> We would appreciate it if you can look over our revised draft and our responses to your review and let us know if you still have other questions.

---

> > ### Comment · Reviewer_s416 · 2024-12-03
> >
> > Thanks to the authors for the additional results and clarifications. I'd like to keep my ratings positive.

---

### Official Review · Reviewer_Ujh5 · 2024-11-03

**Soundness:** 4
**Presentation:** 4
**Contribution:** 2
**Rating:** 3
**Confidence:** 5

**Summary:**

The authors present a paper on a CLIP-based approach combining images, DNA barcodes and textual taxonomic description for biological classification (specifically of insects) in an open-set setting. Experimental results show that the method is effective and outperforms dual-modality contrastive learning approaches, as well as other approaches from the literature on this specific task.

**Strengths:**

1) The paper is well-written and and organized. The proposed approach and the experiments are clearly described.

2) The proposed approach is effective in tackling the problem at hand, while being simpler than common alternatives in the field.

**Weaknesses:**

1) The paper presents no methodological novelty, and mostly applies existing techniques in a standard way to a particular use case.

**Questions:**

1) In Tab. 1, what is the point of comparing non-aligned embeddings? It seems intuitive that there should be no correspondence in the learned representations.

2) The attention visualization is not discussed in detail. Also, as mentioned above, what information comes from checking classification before alignment? Wouldn’t correct predictions be random in that case?

---

> ### Author Response · Authors · 2024-11-20
>
> Thank you for your review. In addition to the general response, we address your specific comments below:
>
> **Methodological novelty**: Please see our general response regarding the novelty of our work. We believe that our experimental findings can benefit downstream use cases and be extended to new problems with similar modality setups.
>
> **Comparing against non-aligned embeddings**: We compare unaligned embeddings for classification against aligned models as a baseline to show the impact of multimodal contrastive learning. The reviewer is correct that in the case of image-to-DNA, as the embeddings are not aligned, the performance is effectively chance level. However, as shown in the other columns of Table 1, classification using DNA-to-DNA and image-to-image performance is fairly high even with the initial encoders, thus demonstrating improvement through cross-modal alignment. Hence we felt it was important to show the performance of the initial models on the intra-modality tasks. While we could mark the cells of this row in the cross-modal (chance) columns as “N/A”, we believe it is more informative to confirm empirically the intuition that unaligned embeddings don’t support cross-modal retrieval.
>
> **Attention visualization**: Our goal with the attention visualization is to analyze how the model’s prediction changes through contrastive training, not just at the prediction level but also in terms of interpretability and feature activation. Since we are visualizing the heatmaps of the image attention outputs on the query image, the unaligned baseline is interpretable and better than chance, even if it wasn’t pretrained on insect data specifically. In the examples shown in figures 5 and 10, we observe that the heatmaps consistently focus more on the full insects for the model trained on all modalities compared to the others, supporting the benefit of cross-modal training to improve the interpretability of the image encoder. We have added additional analysis of the attention maps to Section 5.3 of the paper.

---

> ### Author Response · Authors · 2024-11-27
>
> Thank you for taking the time to review our paper.  Based on suggestions and feedback from you and other reviewers, we have revised our paper to include a unimodal baseline (updated Table 1), experiments with ImageBind-style training (Appendix B.1.1), full taxonomic classification results in our BioCLIP comparison (Appendix B.1.2), additional statistics (Appendix A), a more in-depth discussion of DNA barcodes (Appendix D), as well as other information and clarifications (see general response for details).
>
> Regarding the contribution of our work, we want to emphasize that aside from “methodological novelty”, there are other ways for a work to “contribute new knowledge and sufficient value to the community” (see https://iclr.cc/Conferences/2025/ReviewerGuide), including providing empirical findings.  In our work, we showed that DNA barcodes are a useful signal for aligning image representations, and we contributed a series of experiments to demonstrate this.  We believe that our work can inspire more ML experts from the ICLR community to apply multimodal learning to DNA barcodes, as well as find challenging problems that may require the development of more advanced techniques. We invite you to read the position paper by Rolnick et al. presented earlier in the year at ICML [1]. This paper argues that application-driven research has been systematically under-valued in the ML community. While we do not consider our work to be solely applied, as it involves the novel application of existing ML techniques to a new problem domain and data type, we feel that a dismissal solely on the basis of “no methodological novelty” would be shortsighted and fail to recognize the substantial empirical contribution and potential impact of our work. We believe that our findings represent a significant step forward in the application of machine learning to biological data analysis and that this contribution deserves recognition within the ICLR community.
>
> We would appreciate it if you can look over our revised draft and our responses, consider our contribution to the community, and let us know if you still have questions or concerns.
>
> [1] Rolnick et al. “Position: Application-Driven Innovation in Machine Learning”. ICML (2024).

---

### Author Response · Authors · 2024-11-20
**General Response**

We thank the reviewers for their thoughtful feedback. We appreciate that the reviewers found the problem formulation and motivation interesting (JVaW), especially with the use of DNA (s416, CfWv) and that our proposed approach simply and effectively tackled the problem (Ujh5). The reviewers found our paper and training procedure was well-organized (Ujh5, CfWv, JVaW) and that we had extensive experiments and results for our approach (s416, CfWv).

We provide responses to shared comments below and further direct responses to each respective review:

**Methodological Novelty [Ujh5, JVaW]**: We believe our work provides new value and insights both for biodiversity and for the ML community. For biodiversity, our work demonstrates the possibility of leveraging DNA barcodes for image-based taxonomic classification, significantly improving the workflow of classifying species and allowing for novel downstream tasks that would not be possible without a shared representation space. Such tasks are important for monitoring species-at-risk and evaluating/optimizing conservation efforts. For the ML community, our work is a first yet important step in showing how contrastive learning can leverage modalities of varying information density and scale to improve both unimodal and multimodal retrieval performance. Obtaining human labels is extremely costly in this domain due to the level of expertise required to classify samples to species level. Depending on the sample, it may take a few minutes to half an hour to label a sample from viewing an image, or it may require several hours to examine or dissect the specimen. We propose a solution where a dense modality (barcodes) that can be more easily collected at scale can complement a sparse, human-interpretable modality (taxonomic labels) for training a third modality (vision) ideal for deployment. We anticipate that this problem structure exists elsewhere in ML and our solution generalizes to those settings. Application areas that have such structure are diverse and include medical imaging such as with X-rays or colonoscopies, robotics with tactile sensory data, and key identification in music. We chose to focus specifically on biodiversity because we believe it is itself an important and rich problem and wanted to prioritize depth and exploration of a single problem over shallow coverage of multiple datasets.

**DNA barcode vs taxonomic labels as alignment target [CfWv,JVaW]**: Our experiments show that by aligning to DNA barcodes, instead of taxonomic labels, we learn representations that are better at taxonomic classification (especially at the more challenging genus and species levels). Reviewers CfWv and JVaW had questions about the cost (CfWv) and value (JVaW) of obtaining DNA barcodes vs taxonomic labels. We want to highlight that DNA barcodes can be obtained via a systematic and parallel process while taxonomic labels require manual annotation by scientific experts. In fact, it can be extremely challenging even for experts to classify an insect down to the species level—that is the underlying reason why a majority of the records lack species-level labels. In addition, a specimen’s DNA barcode provides intrinsic identifying markers, while taxonomic labels are designed by humans. The DNA barcodes are a property of the specimen itself, while taxonomic labels are provided by annotators and subject to ambiguity and interpretation. In this work, we are attempting to assert that by aligning image and DNA barcode embeddings, we can train models that do not rely on these costly human-provided taxonomic labels during training. Please also see individual responses for details on cost (CfWv) and challenges of obtaining taxonomic labels (JVaW).

**Additional experiments**: We have improved our manuscript with additional experiments comparing against unimodal contrastive learning (with SimCLR). We included additional results comparing against BioCLIP at higher-level taxa and a chance-level baseline . We have also clarified a few figures and explanations, including more statistics and details about the data used for training and evaluation for each experiment as well as clarified explanations for the attention visualizations. Lastly, we have added Figure 9 to demonstrate the reproducibility of our results across multiple training runs. We mark main changes in magenta.

We thank all of the reviewers for their time in reading our paper and contributing to the discussion.

---

### Author Response · Authors · 2024-11-27
**Summary of Changes**

We thank the reviewers for all of their feedback and comments on our paper.  We have updated the paper with the following:

**Main changes**
1. Experiment comparing initializing CLIBD with pretrained image-encoder on BIOSCAN-1M (Appendix B.1).  Thanks to **R-CfWv** for the suggestion.
   1. We used unimodal contrastive learning (e.g. SimCLR with data augmentation) to first fine-tune a pretrained image-encoder (ViT-B) on BIOSCAN-1M.  Our initial experiments show that using unimodal contrastive learning, we can improve our model performance over the initial ViT-B, but it did not perform as well as our CLIBD multimodal models.
   2. By initializing our CLIBD using this pretrained model (Appendix B.1.1), we obtain higher accuracy in image-to-image and image-to-DNA retrieval at finer taxonomic levels than when initializing with the ViT-B pretrained only on ImageNet.

| Taxa        | SimCLR   | DNA Seen | DNA Unseen | DNA H.M. | Img Seen | Img Unseen | Img H.M. | I-D Seen | I-D Unseen | I-D H.M. |
|-------------|----------|----------|------------|----------|----------|------------|----------|----------|------------|----------|
| Order       | ✗        | **100.0**| **100.0**  | **100.0**| **99.7** | 94.4       | 97.0     | 99.4     | **88.5**   | **93.6** |
|             | ✔️       | **100.0**| **100.0**  | **100.0**| **99.7** | **98.9**   | **99.3** | **99.6** | 75.3       | 84.8     |
| Family      | ✗        | **100.0**| 98.3       | 99.1     | 90.9     | 81.8       | 86.1     | 90.8     | 50.1       | 64.6     |
|             | ✔️       | **100.0**| **99.5**   | **99.7** | **92.2** | **85.8**   | **88.9** | **92.6** | **52.1**   | **66.7** |
| Genus       | ✗        | 98.2     | **94.7**   | **96.4** | 74.6     | 60.4       | 66.8     | 70.6     | 20.8       | 32.1     |
|             | ✔️       | **98.5** | 94.5       | **96.4** | **77.2** | **63.5**   | **69.6** | **73.5** | **23.4**   | **35.5** |
| Species     | ✗        | 95.6     | **90.4**   | **92.9** | 59.3     | 45.0       | 51.2     | 51.6     | 8.6        | 14.7     |
|             | ✔️       | **95.7** | 89.8       | 92.7     | **62.2** | **47.9**   | **54.1** | **56.5** | **9.7**    | **16.5** |

2. Experiment comparing ImageBind style training with frozen BioCLIP image/text encoders (Appendix B.1.1).  Thanks to **R-JVaW** for the suggestion.
   1. We experimented with starting with the BioCLIP image and text encoder and keeping them frozen while aligning the DNA encoder to the image encoder (or image and text encoder together). We found while this was very effective in aligning the DNA encoder, the image encoder performance still lagged behind that of the CLIBD model.

| Taxon. level | Model          | Img     | DNA     | Txt     | I-I Seen | I-I Unseen | I-I H.M. | I-D Seen | I-D Unseen | I-D H.M. | I-T Seen | I-T Unseen | I-T H.M. |
|--------------|----------------|---------|---------|---------|----------|------------|----------|----------|------------|----------|----------|------------|----------|
| Order        | Image          | ❄️       | ✗       | ❄️       | **100.0**| **100.0**  | **100.0**| 88.5     | 86.0       | 87.2     | 87.8     | 64.8       | 74.6     |
|              | Image & Text   | ❄️       | ✗       | ❄️       | **100.0**| **100.0**  | **100.0**| 88.5     | 86.0       | 87.2     | **88.7** | **71.9**   | **79.4** |
| Family       | Image          | ❄️       | ✗       | ❄️       | **100.0**| **97.6**   | **98.8** | 84.3     | 68.4       | 75.5     | **79.6** | 43.1       | 55.9     |
|              | Image & Text   | ❄️       | ✗       | ❄️       | **100.0**| 97.2       | 98.6     | 84.3     | 68.4       | 75.5     | 78.1     | **45.4**   | **57.5** |
| Genus        | Image          | ❄️       | ✗       | ❄️       | 98.2     | 92.9       | 95.5     | 62.6     | 47.1       | 53.7     | 56.0     | 17.9       | 27.1     |
|              | Image & Text   | ❄️       | ✗       | ❄️       | **98.9** | **93.2**   | **96.0** | 62.6     | 47.1       | 53.7     | **56.5** | **19.7**   | **29.2** |
| Species      | Image          | ❄️       | ✗       | ❄️       | 95.1     | **87.9**   | 91.3     | 44.1     | 32.4       | 37.4     | **39.5** | 7.4        | 12.5     |
|              | Image & Text   | ❄️       | ✗       | ❄️       | **95.9** | 88.6       | **92.1** | 44.1     | 32.4       | 37.4     | 38.0     | **8.5**    | **13.9** |

3. Discussion of utility and limitations of using DNA barcodes (Appendix D). This was motivated by **R-JVaW**’s questions concerning cost of acquisition, compute and storage costs of DNA barcodes compared to the costs associated with taxonomic labeling. We also updated the conclusion to acknowledge the limitations of datasets with DNA barcodes and highlight potential future directions.

(cont.)

---

> ### Author Response · Authors · 2024-11-27
>
> (cont.)
>
> 4. Improved description of the BZSL experiment on Insect dataset and its analysis (Page 10). While there were no reviewer comments specifically about this experiment, changes were made regarding concerns by **R-JVaW** about the use of only one dataset in our work and general writing and clarity improvements in this section.
> 5. (From before) Additional data statistics (Appendix A) and full taxonomic results for comparison with BioCLIP (Appendix B.1.2) in response to **R-s416**’s suggestion.
>
> **Other changes**
> 1. Added discussion of related works pointed out by **R-JVaW** (L121-128).
> 2. Improved captions to clarify some figures (Figure 2,4).
> 3. Simplified Figure 2, and added top retrievals to Figure 11 (now expanded into Figures 11-13) to show similarity of retrieved images.
> 4. Other edits to motivate the use of barcodes and clarify our contribution and some methodological and experimental details (see magenta text throughout).
>
> We hope the reviewers will consider our revised draft and update their scores accordingly.  Please let us know if you still have concerns or additional questions.

---

### Author Response · Authors · 2024-11-27
**Summary of Contributions**

**Contributions**

We want to reiterate that our main contribution is using CLIP-style contrastive training to align image and DNA barcode embeddings.  Our experiments on BIOSCAN-1M demonstrate that by aligning to DNA barcodes, we can improve image representations for taxonomic classification.  In addition, by adding sparse taxonomic labels as an alignment target, the performance of our model is further improved.  We also show that DNA barcodes can potentially be a stronger target for alignment than taxonomic labels that are not always available.  This is not necessarily an obvious finding, as shown by our discussion with R-JVaW.

**Stimulate follow up work on using multi-modal learning with DNA barcodes**

Biodiversity monitoring is an important problem.  Recently, there has been an increasing number of datasets (AMI [1], TreeOfLife-10M [2], BioTrove [3]) to engage ML experts to apply ML techniques to assist in biodiversity efforts.  Some are datasets aggregated from different sources (TreeOfLife-10M), while other datasets target a specific workflow (AMI, BIOSCAN).  While our experiments are currently limited to the BIOSCAN-1M and INSECT dataset, we want to emphasize that the BIOSCAN-1M dataset is based on data from active scientists in biodiversity monitoring ([Centre for Biodiversity Genomics](https://biodiversitygenomics.net/)).  It was contributed to help spur investigation of the application of ML techniques on DNA barcodes for various tasks, including taxonomic classification.  In our work, we took the first step in combining information from DNA barcodes, images, and taxonomic labels using CLIP-style contrastive pretraining.  While there are many different multimodal learning and other innovative techniques that can be applied, we believe our work can stimulate further research in this area.

[1] Jain et al. "Insect Identification in the Wild: The AMI Dataset." ECCV 2024.

[2] Stevens et al. "Bioclip: A vision foundation model for the tree of life. "CVPR  2024.

[3] Yang et al. "BioTrove: A Large Curated Image Dataset Enabling AI for Biodiversity." NeurIPS D&B Track  2024.

---

### Meta-Review · Area_Chair_9iGo · 2024-12-21

**Metareview:**

The rebuttal provided clarifications about the proposed method and its analysis that were useful for assessing the paper's contribution and responded adequately to most reviewer concerns. Two reviewers recommend marginal acceptance, and two keep the rejection decision after discussion. The AC agreed this work is valuable to the ICLR community. The final version should include all reviewer comments, suggestions, and additional clarifications from the rebuttal.

**Additional Comments On Reviewer Discussion:**

NA

---

### Decision · Program_Chairs · 2025-01-22

Accept (Poster)